# Piloting the informed health choices resources in Barcelona primary schools: A mixed methods study

**Laura Samsó Jofra**[1,2]*, **Pablo Alonso-Coello**[2,3], **Esther Cánovas Martínez**[2], **Carol de Britos Marsal**[4], **Ana Gallego Iborra**[5], **Ena Pery Niño de Guzman Quispe**[6], **Giordano Pérez-Gaxiola**[7], **Carolina Requeijo**[1], **Marta Roqué i Figuls**[2,3], **Sarah Rosenbaum**[8], **Karla Salas-Gama**[9], **Iratxe Urreta-Barallobre**[10,11], **Laura Martínez García**[2,3]

1 Epidemiology and Public Health Department, Hospital de la Santa Creu i Sant Pau, Barcelona, Spain, 2 Iberoamerican Cochrane Centre (IbCC) - Sant Pau Biomedical Research Institute (IIB-Sant Pau), Barcelona, Spain, 3 CIBER of Epidemiology and Public Health (CIBERESP), Barcelona, Spain, 4 Escola Virolai, Barcelona, Spain, 5 Andalusian Health Service, Malaga, Spain, 6 Cancer Prevention and Control Programme, Catalan Institute of Oncology - IDIBELL, Hospitalet de Llobregat, Barcelona, Spain, 7 Paediatric Hospital of Sinaloa, Culiacan, Mexico, 8 Centre for Epidemic Interventions Research, Norwegian Institute of Public Health, Oslo, Norway, 9 Vall d'Hebron University Hospital, Barcelona, Spain, 10 Clinical Epidemiology, Biodonostia Health Research Institute, San Sebastián, Spain, 11 Clinical Epidemiology Unit, Donostialdea Integrated Health Organisation, Osakidetza Basque Health Service, Donostia University Hospital, San Sebastián, Spain

* lsamso@santpau.cat

**Data Availability Statement:** All relevant data are within the paper and its Supporting information files.

## Abstract

### Introduction

The main objective of the Informed Health Choices (IHC) project is to teach people to assess treatment claims and make informed health choices. For this purpose, the IHC learning resources were developed for primary school children. The aim of this study is to explore students' and teachers' experience when using the IHC resources in primary schools in Barcelona (Spain).

### Methods

We conducted a mixed methods study for piloting the IHC resources in a convenience sample of primary schools in Barcelona. The intervention included a workshop with teachers, and nine lessons with students. We collected data using multiple approaches. We performed quantitative and qualitative analyses, and integrated the findings in a joint display. Finally, we formulated recommendations for using the IHC resources in this setting.

### Results

Two schools, with a total of 143 students in 4th and 5th grade and six teachers, participated in the study. One school followed the suggested IHC teaching plan and competed all the lessons; the other school modified the plan substantially and did not complete all the lessons. Overall, students and teachers from both schools understood, were interested in, and were

**Funding:** This study has been funded by Instituto de Salud Carlos III [CP18/00007] (Cofunded by European Regional Development Fund/European Social Fund, 'Investing in your future'). LMG has a Miguel Servet research contract from the Institute of Health Carlos III [CP18/00007] (Cofunded by European Regional Development Fund/European Social Fund, 'Investing in your future'). Dr. Antoni Esteve Foundation has funded the Spanish translation and production of the IHC resources. The funders had no role in study design, data collection and analysis, decision to publish, or preparation of the manuscript.

**Competing interests:** The authors have declared that no competing interests exist.

able to apply the content of the lessons. During the lessons, the textbook was useful for students; nevertheless, for the teachers, the usefulness of the IHC resources was variable. Teachers adapted the IHC resources to increase student participation and used Information and Communications Technologies tools. We observed more facilitators than barriers to teach the lessons. The teachers suggested some ideas to improve the lessons based on activities they developed and implemented. The integration analysis showed great convergence of the quantitative and qualitative findings. We propose seven recommendations for using the IHC resources in this setting.

## Conclusions

Students and teachers from primary schools in Barcelona showed a positive experience when using IHC resources; however, these resources should be adapted to promote classroom participation.

## Background

Claims about treatments (understanding "treatment" as any action to improve or maintain the health of individuals) are posted by anyone, anywhere, at any time [1]. Treatment claims can be untrustworthy [2, 3]; therefore, it is crucial that people learn to think critically about these claims to make well-informed health choices [4].

### Informed health choices project

The main objective of the Informed Health Choices (IHC) project is to teach people to assess treatment claims and make informed health choices [5]. Initially, the IHC Working Group developed: 1) the IHC Key Concepts (list of concepts that individuals need to understand and apply when assessing claims about treatment effects and making health choices) [6], 2) the IHC learning resources (resources to teach people to understand and apply the IHC key concepts) [5], and 3) the CLAIM Evaluation Tools (item bank with questions to assess people's understanding and ability to apply the IHC key concepts) [7].

Focusing on children, the IHC Working Group developed and tested learning resources for primary school children (10 to 12-year-olds) from a low-income country (Uganda) [8–12]. Currently, they are developing and testing learning resources for secondary school children (14 to 16-year-olds) in low-income countries (Rwanda, Uganda, Kenya) [13].

### Contextualization of the informed health choices resources

The contextualization of the IHC resources involves activities to explore how these resources can be used in a different setting from the one that they were originally designed for (primary schools in Uganda) [14]. The IHC Working Group proposed several contextualization activities 1) context analysis to explore educational conditions in primary schools to teach critical thinking about health, 2) translation of the IHC resources into local language, 3) pilot testing of the IHC resources in primary schools to explore users' experience, 4) adaptation of the IHC resources to improve users' experience (if needed), and 5) translation and validation of the CLAIM Evaluation Tools to measure the ability of primary school children to assess treatment claims and make informed health choices [14].

Different working groups around the world are contextualizing the IHC resources into their settings [15–19]. So far, the main contextualization activities performed include 1)

translation of the IHC resources (available on the IHC website in 13 languages) [5], 2) pilot testing of the IHC resources in primary schools [20–23], and 3) translation and validation of the CLAIM Evaluation Tools [24–26].

Currently, there are no specific learning resources to teach primary school children to think critically about their health in Spanish context. The IHC-Barcelona Working Group translated the IHC resources into Spanish [27–29]. The next step was to pilot test these resources and ensure their appropriateness for Spanish primary school children. For this purpose, we conducted this mixed methods study aimed to 1) explore the students' and teachers' experience when using the IHC primary school resources in Barcelona (Spain), and 2) formulate recommendations to use the IHC primary school resources in this setting.

## Methods

### Design

We conducted a convergent mixed methods study [30]. We used multiple approaches to collect in parallel quantitative data (teachers' questionnaires) and qualitative data (lessons' non-participatory observations [NPOs], and students' semi-structured interviews [SSIs]) on users' experience with the IHC resources (S1 File). The quantitative and qualitative data were collected, analysed, and interpreted separately. Finally, we integrated the qualitative and quantitative findings for an in-depth understanding of the same phenomenon (triangulation purpose) [31].

An extended description of methods is available in a previously published protocol [32]. We used the Good Reporting of A Mixed Methods Study (GRAMMS) checklist to report the paper (S2 File) [33].

### Setting

The study was conducted in Barcelona (Spain). Barcelona is an urban area with ten districts, 73 neighbourhoods, and around 1,600,00 habitants [34]. For 2019–2020 school year, the statistical yearbook of the city included 344 primary schools (6 to 12-year-olds); of which 173 were public and 171 were publicly-funded private or private schools [34]. The number of students and teachers reported was around 82,000 and 3,800 respectively [34].

**Establishment of the IHC-Barcelona working group.** The IHC-Barcelona Working Group included a coordination team (which led and coordinated the study), and a multidisciplinary Advisory Group (which reviewed and advised on the different steps of the study). We aimed for profile representativeness of the Working Group members (researchers, teachers, family members, paediatric primary care providers, and education and health stakeholders).

**Selection of schools.** We selected a convenience sample of three schools in Barcelona. We used the following eligibility criteria: 1) schools included in the school directory from the Regional Ministry of Education from the Government of Catalonia (2018–2019) [35]; 2) schools that had previously participated in a health promotion programme (2016–2017) [36]; and 3) schools that had previously participated in the initiative *Escola Nova 21* (alliance of schools and civil society institutions for updating the Catalan education system during 2016–2019) [37]. We also took into consideration whether the schools included students that were representative of the neighbourhood, if they were in different neighbourhoods of the city, their type of funding (public, publicly-funded private, or private schools), and schools that previously participated in the IHC resources translation into Spanish. Some of these criteria were established to identify schools that had previously collaborated on related initiatives (health promotion programmes, initiative Escola Nova 21, or the IHC project), and thus guarantee participation in the study and intervention adherence.

**Selection of the students and teachers.** We included 4th and 5th-grade students (9 to 11-year-olds) from all the classes of the participating schools, and one teacher from each included class.

**Intervention in the schools.** We scheduled the intervention for 2019–2020 school year (first intervention or intervention pre COVID-19 pandemic), but in mid-March 2020 all schools closed due to the COVID-19 pandemic. We finished the data collection of the first intervention during the COVID-19 pandemic lockdown. We restarted the intervention in the 2020–2021 school year (second intervention or intervention after COVID-19 pandemic lockdown) and adapted the methods to schools' COVID-19 prevention and control measures.

The intervention in the schools included: 1) a workshop with the teachers, 2) lessons taught to the students, and 3) assessment of CLAIMs about treatments by the students at the end of the lessons. Each of the activities is summarised below:

- **Workshop with the teachers:** Before starting the lessons, we ran a workshop to introduce and review the IHC project, the IHC resources, and the study with the teachers. The workshop program included the following sessions: 1) an opening session about Evidence-Based Medicine, 2) the presentation of the IHC project and the IHC resources, 3) the presentation of the study, 4) a mock lesson to the teachers, and 5) teachers' presentation about their plan to teach the lessons to the students (the workshop program is available in S3 File). The workshop lasted five hours.

- **Lessons to the students:** The IHC primary school resources include a textbook, an exercise book, a teachers' guide, some activity cards, a poster, and a song (Fig 1) [10–12, 27–29]. The textbook includes nine lessons that explain 12 IHC key concepts (S4 File). The textbook tells a story, narrated as a comic, about a brother and a sister, John and Julie, who meet two teachers and health researchers, professor Compare and professor Fair. The professors teach the children: 1) what questions they should ask when someone says something about a

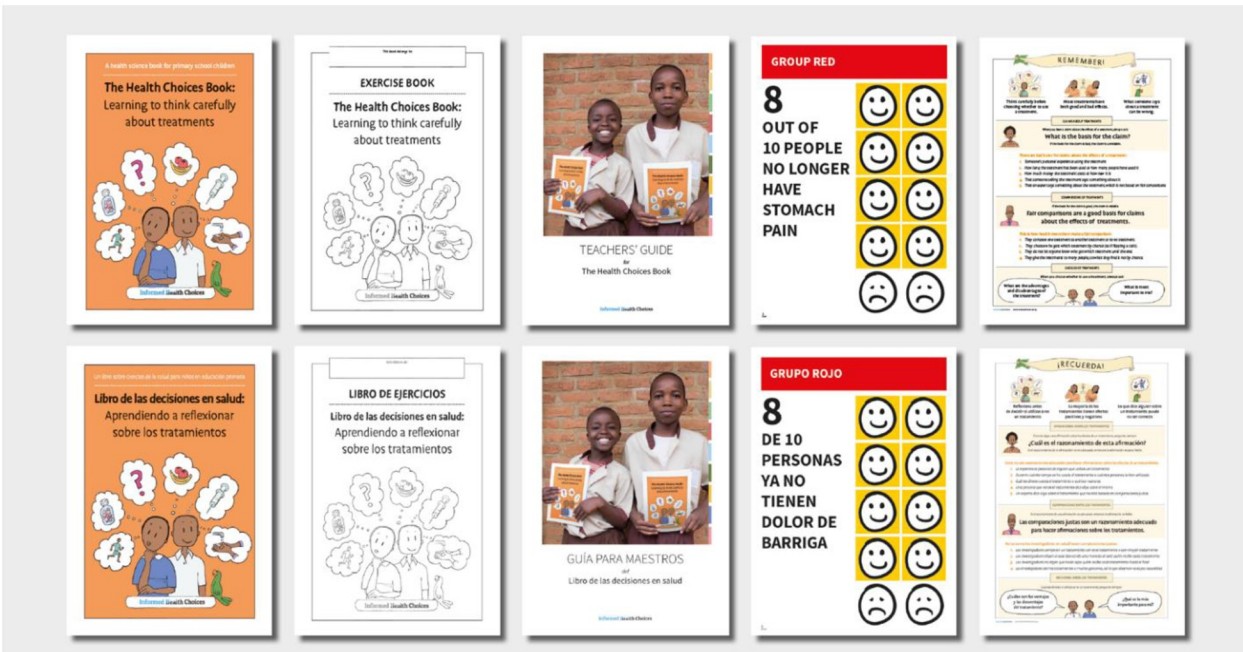

**Fig 1. The informed health choices learning resources for primary school children (English and Spanish version)** [10–12, 27–29].

treatment; 2) what questions health researchers ask to find out more about treatment effects; and 3) what questions they should ask when deciding to use a treatment or not. The IHC resources were designed to be managed by teachers, used over nine weeks, with one lesson per week, during a single term, and one hour to complete a test at the end of the term [9].

We provided schools with the IHC resources. Each teacher received the following printed copies: a textbook, a teacher's guide, a packet of activity cards and a poster for the class. Each student, if requested by the teacher, received a printed copy of the textbook. We did not provide printed copies of the exercise book for students, instead we asked the teachers to print the exercises if needed.

We asked teachers to at least read and discuss the story for each lesson. Beyond that, they could adapt the lessons to their students depending on the education plan of each school (e.g., continuity and duration of the lessons, completion of the activities/exercises, development of extra activities, resource format and language).

- **Assessment of CLAIMs about treatments by the students at the end of the lessons (CLAIM test):** After completing all the lessons, the students completed a self-administered test (CLAIM test) to evaluate their ability to apply the 12 IHC key concepts taught during the lessons. The test included 24 questions (two questions for each IHC key concept) from the CLAIM Evaluation Tools item bank [7]. During the first intervention (and under the COVID-19 pandemic lockdown), the students completed the CLAIM test at home and sent it by email. During the second intervention, the students completed the CLAIM test during school time.

## Participants

**Quantitative sample.** All the teachers who participated in the intervention completed *ad hoc* self-administered questionnaires (before the lessons, after each lesson, and at the end of the lessons) to explore the users' experience with the IHC resources.

**Qualitative sample.** We selected a convenience sample of lessons to conduct non-participatory observations (NPOs) during the lessons and semi-structured interviews (SSIs) with students after the lessons. We expected to perform at least two NPOs from each lesson in randomly assigned classes (18 observations) and two SSIs of each lesson on randomly assigned students (18 interviews).

## Data collection

**Quantitative data.**

- **Assessment of the IHC resources by the teachers before the lessons:** Before staring the lessons, the teachers completed an *ad hoc* self-administered questionnaire that assessed: 1) teacher's impression of understandability (easy for users to comprehend), desirability (something the users like), suitability (something the users feel is for "someone like me"), and usefulness (helpful to users in achieving their needs) of the IHC resources for their students [38]; 2) understandability, desirability, suitability, and usefulness for themselves; and 3) examples of treatment claims (open-ended questions) (S5.1 File in S5 File). In this assessment, and in the following ones, the understandability, desirability, suitability, and usefulness of the IHC resources were evaluated using a 5-Point Scale (1 meaning completely disagree and 5 meaning completely agree).

- **Assessment of the lessons by the teachers after a lesson:** After teaching each lesson, the teachers completed an *ad hoc* self-administered questionnaire that assessed: 1) teacher's impression of understandability, desirability, suitability, and usefulness for students; 2) understandability, desirability, suitability, and usefulness for themselves; 3) technique used to teach the lessons (close and open-ended questions), 4) facilitators and barriers to teach the lessons, and 5) suggestions to improve the lessons (open-ended questions) (S5.2 File in S5 File).

- **Overall assessment of the IHC resources by the teachers at the end of the lessons:** After completing all the lessons, the teachers completed an *ad hoc* self-administered questionnaire that assessed: 1) teacher's impression of overall understandability, desirability, suitability, and usefulness for students; 2) overall understandability, desirability, suitability, and usefulness for themselves; and 3) suggestions to improve the IHC resources (open-ended questions) (S5.3 File in S5 File).

   **Qualitative data.**

- **Non-participatory observations during the lessons (NPOs):** Two researchers from the IHC-Barcelona Working Group made the NPOs during the lessons (NPOs were audio-recorded). The researchers conducted the NPOs using an *ad hoc* guide that assessed: 1) understandability, desirability, suitability, and usefulness for students and teachers, 2) technique used to teach the lessons, 3) facilitators and barriers to teach the lessons, and 4) examples of treatment claims (S5.4 File in S5 File). The researchers made detailed notes during NPOs; these were collected as direct quotes, paraphrases, or researcher hypotheses. They cross-checked quotes and solved discrepancies through discussion; if necessary, they reviewed the audio-recordings. During the second intervention, due to COVID-19 restrictions, we held the NPOs on-line.

- **Semi-structured interviews with the students after a lesson (SSIs):** After each NPO, two researchers from the IHC-Barcelona Working Group made the SSIs (SSIs were also audio-recorded). The researchers conducted the SSIs using an *ad hoc* guide that assessed: 1) understandability, desirability, suitability, and usefulness for students, 2) examples of treatment claims, and 3) suggestions to improve the lessons (S5.5 File in S5 File). The researchers made detailed notes during SSIs; these were collected as direct quotes, paraphrases, or researcher hypotheses. They cross-checked quotes and solved discrepancies through discussion; if necessary, they reviewed the audio-recordings. During the second intervention, due to COVID-19 restrictions, we held the SSIs on-line.

## Data analysis

**Quantitative analysis.**   We conducted a descriptive analysis of the categorical variables (absolute and relative frequencies), and the continuous variables (median and range) (S1 File). We only considered the CLAIM test results for the students at schools that completed all the lessons. We calculated the proportion of students with a passing score (basic knowledge of the concepts and how to apply them, 13 points or more over 24), and the proportion of the students with a mastery score (clear knowledge of the concepts and how to apply them, 20 points or more over 24) [39].

**Qualitative analysis.**   We analysed qualitative data derived from lessons' NPOs, students' SSIs, and free-text responses of teachers' questionnaires (S1 File). We conducted a framework deductive analysis [40] for qualitative data related to several domains of an adapted

version of a user-experience honeycomb framework: understandability, desirability, suitability, and usefulness [38]. We applied the following steps: 1) categorisation of quotes using the framework's themes, and 2) proposal of subthemes under the themes. We conducted a thematic inductive analysis for qualitative data not suitable for the framework: technique used to teach the lessons, facilitators and barriers to teach the lessons, examples of claims about treatment effects, and suggestions to improve the lessons. We applied the following steps: 1) codification of quotes, 2) proposal of descriptive themes, and 3) identification of analytic themes. One researcher categorised and coded quotes, and proposed themes and subthemes independently. A second researcher cross-checked codes, corresponding quotes, themes, and subthemes. Disagreements were initially solved by consensus; if necessary, a third reviewer was consulted.

**Mixed methods integration analysis.** We merged quantitative and qualitative results using a joint display to compare and validate the findings [41]. We applied the following steps: 1) mapping quantitative and qualitative results by outcome into a summarized matrix, 2) exploring the convergence (findings from quantitative and qualitative approach agree), complementarity (findings from each approach offer complementary information), or discrepancy (findings from each approach appear to be contradictive) [42], and 3) narrative synthesising of the integration findings. One reviewer conducted the integration analysis independently. A second reviewer cross-checked the summarised results matrix and the narrative synthesis. Disagreements were initially solved by consensus; if necessary, a third reviewer was consulted.

### Formulation of the recommendations

The IHC-Barcelona Working Group discussed the findings, its relevance for their setting, and its implications for changes to the IHC resources [8]; finally, they agreed on recommendations to improve the users' experience with the IHC resources.

### User participation

Representatives from all the different areas of interest (researchers, teachers, paediatricians, student representatives, family representatives, and education and health stakeholders) were invited to be members of the IHC-Barcelona Working Group.

### Ethical considerations

The study protocol obtained an approval exemption (it did not include patients, biological specimens or clinical data) from the Ethics Committee of the Hospital de la Santa Creu i Sant Pau (Barcelona, Spain) [43]. We informed all participants and students' families about the study, and we requested their written informed consent. If a family did not want to participate, the student participated in the lessons as a curricular activity but did not participate in any of the data collection activities. We also anonymised the data removing participant and school names.

## Results

### Intervention in the schools

**Participants.** The IHC-Barcelona Working Group included 21 participants (researchers, teachers, family members, and paediatric primary care providers) (Table 1), most of them were women (18/21; 85.7%).

**Table 1. Participants of the study.**

| | 1st intervention (2019–2020 school year) | 2nd intervention (2020–2021 school year) | Overall |
|---|---|---|---|
| **Members of the IHC-Barcelona Working Group, n (%)** | | | |
| • Researchers | N/A | N/A | 11 (52.4) |
| • Teachers | N/A | N/A | 5 (23.8) |
| • Family members | N/A | N/A | 3 (14.3) |
| • Paediatric primary care providers | N/A | N/A | 2 (9.5) |
| **Total** | **N/A** | **N/A** | **21 (100)** |
| **Students per school, n (%)** | | | |
| • School A | 51 (51.0) | N/A | 51 (35.7) |
| • School B | 49 (49.0) | 43 (100) | 92 (64.3) |
| **Total** | **100 (100)** | **43 (100)** | **143 (100)** |
| **Students per grade, n (%)** | | | |
| • Class A 5th grade School A (10 to 11-year-olds) | 26 (26.0) | N/A | 26 (18.2) |
| • Class B 5th grade School A (10 to 11-year-olds) | 25 (25.0) | N/A | 25 (17.5) |
| • Class A 5th grade School B (10 to 11-year-olds) | 25 (25.0) | N/A | 25 (17.5) |
| • Class B 5th grade School B (10 to 11-year-olds) | 24 (24.0) | N/A | 24 (16.8) |
| • Class A 4-5th grade School B (9 to 11-year-olds) [a] | N/A | 22 (51.2) | 22 (15.4) |
| • Class B 4-5th grade School B (9 to 11-year-olds) [a] | N/A | 21 (48.8) | 21 (14.7) |
| **Total** | **100 (100)** | **43 (100)** | **143 (100)** |
| **Teachers per school, n (%)** | | | |
| • School A | 2 (50.0) | N/A | 2 (33.3) |
| • School B [b] | 2 (50.0) | 2 (100) | 4 (66.6) |
| **Total** | **4 (100)** | **2 (100)** | **6 (100)** |
| **Informed consent from students' families, n/N [c] (%)** | | | |
| • AGREE to participate in the study | 100/103 (97.1) | 43/44 (97.7) | 143/147 (97.3) |
| **Participants in the workshop [d], n (%)** | | | |
| • Researchers | 8 (53.3) | N/A | N/A |
| • Teachers | 4 (26.7) | N/A | N/A |
| • Family members | 3 (20.0) | N/A | N/A |
| **Total** | **15 (100)** | **N/A** | **N/A** |
| **Students who participated in the CLAIM Test per school, n/N [f] (%)** | | | |
| • School A | 37/51 (72.5) | N/A | N/A |
| • School B [e] | N/A | N/A | N/A |

Abbreviations: IHC: Informed Health Choices; N/A: Not applicable.

[a] During the second intervention, the school reduced the number of students per class (as a COVID-19 prevention and control measure) by: 1) increasing the number of classes, 2) mixing 4th and 5th-grade students in the same class.

[b] One teacher participated in 1st and 2nd intervention.

[c] n (students agreed to participate) / N (students invited to participate).

[d] Participants in the workshop were part of the IHC-Barcelona Working Group.

[f] n (students who completed the CLAIM questionnaire test) / N (students who completed all the lessons).

[e] School B did not complete all the lessons.

We identified 947 schools in Barcelona [35]; of those, 34 fulfilled the eligibility criteria and three agreed to participate in the study (School A, B, and C) (S6 File). Two primary schools in Barcelona (one public [School A], and one publicly-funded private school [School B]) conducted the intervention. The school that dropped out (School C) was a public school,

included a more disadvantaged population, younger students (9 to 10-years-old), only one teacher agreed to participate, and planned to include the lessons in the "Spanish language" subject.

We included a total of 143 students in the study, 100 during the first intervention (pre COVID-19 pandemic) and 43 during the second intervention (after COVID-19 pandemic lockdown) (Table 1). A higher proportion of students were 10 to 11-year-olds (100/143; 69.9%), and there were more boys (81/143; 56,6%) than girls. The students were divided in six classes, with 21 to 26 students in each class. Six teachers participated in the study; all of them were women. The teachers presented the study to the family members during school meetings, and most families agreed to participate (143/147; 97.3%).

**Workshop with the teachers.**   The teachers and other members of the IHC-Barcelona Working Group participated in the workshop (15 participants) (Table 1). The workshop took place in November 2019 at Hospital de la Santa Creu i Sant Pau (Barcelona).

**Lessons to the students.**   One school (School A) completed all the lessons during the first intervention (January-March 2020) (S7.1 File in S7 File). Each teacher from School A taught the lessons to one class. They scheduled one lesson *per* week, with approximately 60 minutes *per* lesson. The lessons were included in two different subjects, "Social and civic values" and "Spanish language", and were conducted in Spanish. They used printed IHC resources translated into Spanish, and we provided a printed copy of the textbook for each student.

The other school (School B) only completed two lessons during the first intervention (February-March 2020), and 3–5 lessons during the second intervention (April-June 2021) (S7.1 File in S7 File). Each teacher from School B taught the lessons to one class. They adapted the lessons to include them in their extensive project-based learning methodology (each lesson was divided into different sessions to work together with other related content in the curriculum) [44]. The lessons were included in the subject "Science" and were conducted in English as it is the vehicular language of this subject. They used digital IHC resources in English, and we provided some printed copies of the textbook translated into Spanish for students with English comprehension difficulties.

**CLAIM test.**   A total of 37 students completed the CLAIM test (37/51 students from School A that completed all the lessons, response rate 72.5%, Table 1). Almost all the students obtained a passing score on the test (36/37, 97.3%), and more than half of the students achieved a mastery score (23/37, 62.2%) (S7.2 File in S7 File).

## Students' and teachers' experience with the IHC resources

**Quantitative findings.**   We collected quantitative data from the following data collection activities (S7.1 File in S7 File): six assessments of the IHC resources by the teachers before the lessons, 26 assessments of the lessons by the teachers after a lesson, and two overall assessments of the IHC resources by the teachers at the end of the lessons.

*Understandability, desirability, suitability, and usefulness of the IHC resources.* Before starting the lessons, teachers expected that students would understand and would be able to apply the content of the lessons in their daily lives (median score 4 [1 meaning completely disagree and 5 meaning completely agree] in assessment before the lessons for both items), but they thought that students would be slightly less interested in them (median score 3.5 in assessment before the lessons) (Table 2). During the lessons, students understood (median score 5 in assessment after a lesson), were interested (median score 4 in assessment after a lesson), and were able to apply the content of the lessons (median score 4 in assessment after a lesson). Teachers understood the content of the IHC resources (median score 5 in all the assessments), found them interesting (median score 4.5 in assessment before the lessons, and median score 5

**Table 2. Students' and teachers' experience with the IHC resources.**

| | Students' experience, median (range) [a], colour scale [b] | Teachers' experience, median (range) [a], colour scale [b] |
|---|---|---|
| **Assessment of the IHC resources by the teachers before the lessons (n = 6)** | | |
| Understandability | 4 (4–5) | 5 (4–5) |
| Desirability | 3.5 (3–5) | 4.5 (4–5) |
| Suitability | 4 (3–4) | 4 (3–5) |
| Usefulness—Overall | 4 (2–5) | N/A |
| Usefulness—By resource | | |
| • Health Choices Book | N/A | 4.5 (4–5) |
| • Teachers' Guide | N/A | 3.5 (2–5) |
| • Activity Cards [c] | N/A | 3.5 (2–5) |
| • Poster | N/A | 3 (3–4) |
| **Assessment of the lessons by the teachers after a lesson (n = 26)** | | |
| Understandability | 5 (4–5) | 5 (5–5) |
| Desirability | 4 (3–5) | 5 (3–5) |
| Suitability | 4 (3–5) | 5 (4–5) |
| Usefulness—By resource | | |
| • Health Choices Book | 5 (1–5) | 5 (1–5) |
| • Teachers' Guide | N/A | 5 (1–5) |
| • Activity Cards [c] | 5 (5–5) | 5 (5–5) |
| • Poster | 1 (1–5) | 1 (1–4) |
| **Overall assessment of the IHC resources by the teachers at the end of the lessons (n = 2) [d]** | | |
| Understandability | 4.5 (4–5) | 5 (5–5) |
| Desirability | 4.5 (4–5) | 5 (5–5) |
| Suitability | 4.5 (4–5) | 4.5 (4–5) |
| Usefulness—Overall | 5 (5–5) | N/A |
| Usefulness—By resource | | |
| • Health Choices Book | N/A | 5 (5–5) |
| • Teachers' Guide | N/A | 5 (5–5) |
| • Activity Cards [c] | N/A | 4 (4–4) |
| • Poster | N/A | 2 (1–3) |

Abbreviations: IHC: Informed Health Choices; N/A: Not applicable.

[a] 5-Point Scale (1 meaning completely disagree and 5 meaning completely agree).

[b] Colour scale for median and range.

[c] Only applicable in lesson 7.

[d] Teachers from School A that completed all the lessons.

in assessment after a lesson and at the end of the lessons) and suitable (median score 4 in assessment before the lessons, 5 in assessment after a lesson, and 4.5 at the end of the lessons).

In terms of usefulness of the IHC resources, teachers thought that the IHC resources would be useful for the students (median score 4 in assessment before the lessons, 5 for the textbook in assessment after a lesson, and 5 at the end of the lessons). Before starting the lessons, the teachers mainly expected the textbook to be useful (median score 4.5 in assessment before the lessons); after the lessons and at the end of the lessons, they reported that both, the textbook and the teachers' guide, were useful (median score 5 for the textbook and the teachers' guide in both assessments). Teachers found the poster not useful, neither for the students nor for themselves.

*Technique used to teach the lessons.* The teachers used the techniques of the proposed plan in the IHC resources to teach the lessons, including a review of the previous lesson, reading

the lesson's comic, a discussion, and the completion of the activity/exercises (S7.3 File in S7 File). They also used other methods or strategies.

*Facilitators and barriers to teach the lessons*. The most frequent facilitators for teaching the lessons that teachers identified were related to themselves (their ability to adapt the instructions to their own style and context, their feeling of self-efficacy, and their understanding of the content), the IHC resources (credibility of the material), and the students (attitudes, beliefs, and motivation to learn) (Table 3).

Teachers identified fewer barriers than facilitators (Table 3). Most of them were related to the students (peer influence and differentiated instruction) and the school system and environment (time constraints).

**Qualitative findings.** We collected qualitative data from the following data collection activities (S7.1 File in S7 File): 14 NPOs and 14 SSIs (nine on-site during the first intervention and five on-line during the second intervention, one or two observations/interviews for each lesson, except for Lesson 6 which we did not attend). We also added the answers from open-ended questions included in the questionnaires. Two researchers from the Working Group conducted the NPOs and SSIs (both were female, medical doctors and one of them had experience in qualitative research). A higher proportion of interviewed students were 10 to 11-year-olds (9/14; 64.3%), and the majority were girls (8/14; 57.61%).

*Understandability, desirability, suitability, and usefulness of the IHC resources*. During the lessons, students and teachers understood, were interested, and were able to apply the content of the IHC resources. Relevant quotes about the understanding of the content were noted (Fig 2). Students defined some of the terms to work on (e.g., definition of health or treatment) (Fig 2, quotes 1 and 2). They also addressed other concepts not included in the IHC resources (e.g., placebo effect, washout period, or cumulative effect) (Fig 2, quotes 3–5). Moreover, they established relationships between different concepts (e.g., unreliable claims could be based on more than one bad basis, or positive effects *versus* disadvantages of a treatment) (Fig 2, quotes 6 and 7). Teachers also reinforced the meaning of the concepts by repeating their definition, giving examples, or transferring the concepts to other areas (Fig 2, quotes 8–10). In terms of usefulness of the IHC resources, we observed that students used the textbook (printed or digital format); however, teachers rarely used their printed copy of the textbook or teachers' guide during the lessons.

All the interviewed students were able to explain what they had learned after the lessons (Fig 2, quote 11–13). A few students identified some concepts that were difficult to understand (e.g., random allocation or how to make a decision about treatments) (Fig 2, quote 14 and 15). Most of the interviewed students enjoyed the lessons and thought they were interesting. Some of them explained the aspects of the lessons that they enjoyed the most (e.g., learning something new, the comic, and the animated version of Lesson 2) or disliked the most (e.g., explanations of concepts and repetitive activities) (Fig 2, quote 16–20). More than half of the students expressed that they could apply what they learned in their daily life; when we asked about the reason, they were able to identify their treatments (e.g., vaccines, bandages, or ibuprofen) and the advantages and disadvantages of those (Fig 2, quote 21). Moreover, some of them considered that what they learned could be helpful for their family (Fig 2, quote 22). Only one of the interviewed students noted that the textbook included examples from other settings (Fig 2, quote 23).

*Technique used to teach the lessons*. Teachers from School A almost always followed the proposed plan in the IHC resources, including a review of the previous lesson, reading the lesson's comic, a discussion, and the completion of the activity/exercises. Teachers from the School B implemented the lessons as project-based learning, and they substantially changed the proposed plan in the IHC resources (S7.4 File in S7 File). Nevertheless, they maintained some

**Table 3. Facilitators and barriers to teach the lessons.**

| Facilitators | Definition [45] | n (%) [a] |
|---|---|---|
| **Teachers** | | |
| • Fit to the teacher's teaching style and context | Teachers' comfort or ability to adapt the instructions to their style and context | 26 (100) |
| • Self-efficacy | Teacher's confidence in teaching the lessons | 26 (100) |
| • Understanding of the content being taught | Teachers' understanding of the context | 26 (100) |
| • Positive learning environment | Teachers' ability to create a positive learning environment; for example, encourage discussion, respond positively to questions, engage students | 18 (69.2) |
| • Profiles and competences | Teacher's education and experience in relation to the lessons being taught | 17 (65.4) |
| • Sufficient training | The extent to which the teachers received sufficient training in teaching the lessons | 17 (65.4) |
| • Attitudes | Teachers' attitude towards new resources (change), science, critical thinking and independent thinking by the student body (or their role as authorities in the classroom) | 8 (30.8) |
| • Emotions | Teachers' emotions, such as stress or anxiety | 8 (30.8) |
| **Students** | | |
| • Attitudes | Students' attendance or reasons for poor attendance (e.g., long distance to school or inability to pay school fees) | 16 (61.5) |
| • Beliefs | Students' beliefs about the content (e.g., what treatments work or the concepts) | 15 (57.5) |
| • Motivation to learn | Students' motivation to learn the new material | 15 (57.5) |
| • Attendance | Students' attendance or reasons for poor attendance (e.g., long distance to school or inability to pay school fees) | 8 (30.8) |
| • Peer influence | Positive or negative attitudes of other students towards the material | 4 (15.4) |
| • Literacy | Students' ability to read and understand the material | 3 (11.4) |
| **Learning resources** | | |
| • Credibility of the material | The extent to which the teachers and students perceive the resources as credible | 26 (100) |
| • Compatibility with the curriculum | The extent to which the resources fit with the rest of the curriculum and how it is taught | 8 (30.8) |
| • Appropriateness of the material | The extent to which the resources are relevant, challenging and engaging | 4 (15.4) |
| • Value of the material | The extent to which the materials are valued by the teachers and students | 4 (15.4) |
| **School system and environment** | | |
| • Parent and community involvement | Parents' attitudes towards the new resources or how things are done at the school | 9 (34.6) |
| • Attitudes and beliefs of head teacher and other teachers | Attitudes or beliefs of colleagues that influence the teacher's interest in and ability to teach the material | 8 (30.8) |
| • School organisation and management | The extent to which the school provides an environment that supports adoption of new subjects, resources and teaching methods | 8 (30.8) |
| **Barriers** | **Definition [45]** | **n (%) [a]** |
| **Teachers** | | |
| • Sufficient training | The extent to which the teachers received sufficient training in teaching the lessons | 2 (7.7) |

(*Continued*)

**Table 3.** (Continued)

| | | |
|---|---|---|
| • Emotions | Teachers' emotions, such as stress or anxiety | 1 (3.8) |
| **Students** | | |
| • Peer influence | Positive or negative attitudes of other students towards the material | 11 (42.3) |
| • Differentiated instruction | The extent to which students different learning needs are met | 5 (19.2) |
| • Attitudes | Students' attitudes towards learning, towards authorities, towards science, towards critical thinking | 1 (3.8) |
| • Motivation to learn | Students' motivation to learn the new material | 1 (3.8) |
| **Learning resources** | | |
| • Appropriateness of the material | The extent to which the resources are relevant, challenging and engaging | 3 (11.5) |
| **School system and environment** | | |
| • Time constraints | The extent to which there is sufficient time to accommodate introducing the new material | 5 (19.2) |

This table has been adapted with permission from Nsangi et al. [45].

[a] n (%): times the facilitator/barrier has been reported by teachers in a total of 26 assessments after lessons.

steps of the plan, such as reading the lesson's comic, the discussion, and completion of exercises.

In both schools, teachers used other methods or strategies to teach the lessons, which we classified into two main themes: adaptation of the IHC resources, and use of Information and Communications Technologies (ICT) tools.

- **Adaptation of the IHC resources:** All teachers developed other activities/exercises and work materials based on the IHC resources to promote students' participation. Some of the resources developed by teachers from School A included: a worksheet with concepts and definitions (Fig 2, quote 24; S8.1 File in S8 File), cards with the treatment illustrations from the textbook (Fig 2, quote 25; S8.2 File in S8 File), a diagram with treatment advantages and disadvantages (Fig 2, quote 26; S8.3 File in S8 File), green/red response cards for true or false exercises (Fig 2, quote 27), or a summary activity (Fig 2, quote 28).

Teachers from School B also developed resources to work on IHC contents, for example, a worksheet with examples of claims (S8.4 File in S8 File), a worksheet with concepts and definitions (S8.5 File in S8 File), an exercise to identify bad bases for claims (Fig 2, quote 29; S8.6 File in S8 File), and a worksheet with exercises (S8.7 File in S8 File). Furthermore, they developed resources *ad hoc* to work on other related content, including how to develop and analyse surveys (health survey activity [Fig 2, quote 30; S8.8 File in S8 File]), the comic format (a questionnaire about the comic [S8.9 File in S8 File], a worksheet to create a new comic character [S8.10 File in S8 File]), or English language (English vocabulary exercise about common symptoms [S8.11 File in S8 File], a worksheet to develop a health word search [S8.12 File in S8 File], a reading comprehension worksheet [S8.13 File in S8 File], and an exercise to create or listen to dialogues between doctors and patients [S8.14 and S8.15 Files in S8 File]).

- **Use of Information and Communications Technologies (ICT) tools:** Teachers and students used different ICT tools (including equipment [hardware] and computer programs [software]) during the lessons. For example, teachers from School A created PowerPoint presentations for each lesson to explain the content and present the activities/exercises (Fig 2,

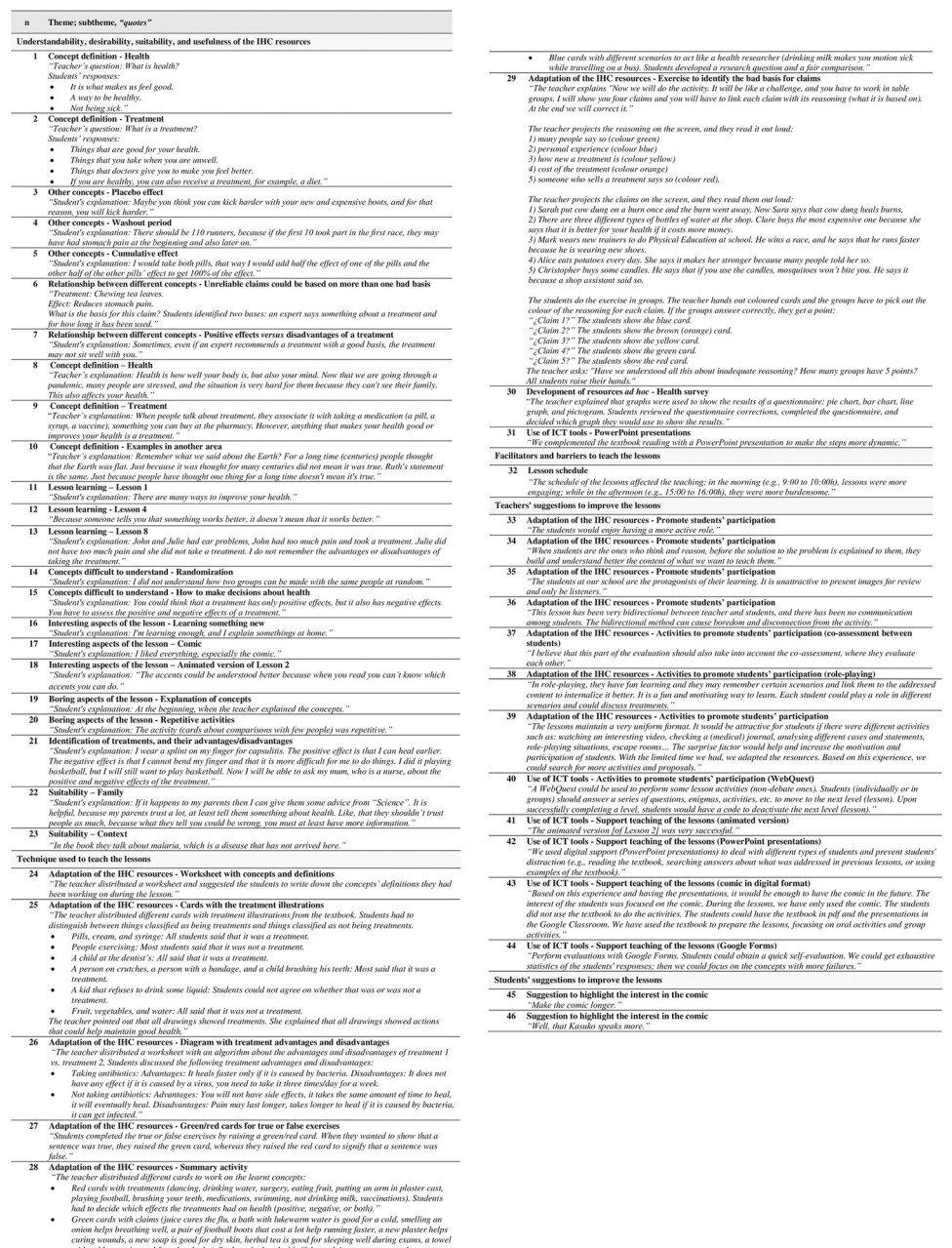

**Fig 2. Relevant quotes.** Abbreviations: ICT: Information and Communications Technologies; IHC: Informed Health Choices.

quote 31). Teachers from School B projected the comic on a screen to read the comic aloud, and students from School B had personal computers and they used the application Google Classroom to download the textbook and lesson materials.

*Facilitators and barriers to teach the lessons*. Teachers were deeply engaged to teach the lessons (attitudes, fit to the teacher's teaching style and context, self-efficacy, motivation, and positive learning environment). Students also contributed to the adequate progress of the lessons (literacy, motivation to learn, peer influence, and attitudes).

We identified few barriers, some of these were related to the students (peer influence and attitudes) and the school system and environment (the schedule of the lessons affected the teaching, as the students' attention was lower in the afternoon lessons than in the morning ones [Fig 2, quote 32]).

We observed circumstances that were both, a facilitator and a barrier. For example, sometimes students' attitude and peer influence facilitated the classroom dynamics (e.g., a student asked a question, and a classmate answered it), but other times they hindered the progress of the lesson (e.g., a noisy classroom).

*Examples of claims about treatment effects.* During the first intervention, students and teachers identified examples from their daily lives, mainly based on personal experiences and non-serious illnesses (e.g., bites, colds, specific pain, or aesthetic treatments) (Fig 3). During the second intervention, students and teachers gave several examples of treatment claims related to COVID-19.

*Suggestions to improve the lessons.* Teachers suggested ideas to improve the lessons based on the activities that they implemented to teach the lessons or their teaching experience. We used the two themes above to classify the suggestions: 1) adaptation of the IHC resources to promote students' participation (Fig 2, quotes 33–39), and 2) use of ICT tools to promote students' participation (Fig 2, quote 40) or to support the teaching of the lessons (Fig 2, quotes 41–44). Some of the interviewed students also made suggestions to improve the lessons and highlighted that the comic was what interested them the most (Fig 2; quotes 45 and 46).

**Mixed methods integration findings.** When integrating the qualitative and quantitative findings into a summarized matrix, we observed almost a complete convergence of the results (S7.5 File in S7 File). Although, two integration concepts emerged:

| **Claims about treatments for a bite** |
|---|
| Hot water to reduce pain from spider fish sting |
| Urine cures jellyfish stings |
| Mud or vinegar decrease nettle sting |
| **Claims about treatments for a cold** |
| Ginger tea with honey and lemon improves sore throat |
| Sleeping with a cut raw onion next to your bed improves your breathing when you have a cold |
| Syrup to prevent sneezing |
| **Claims about treatments for a specific pain** |
| Applying ice on a welt reduces pain |
| Cotton impregnated with olive oil improves ear pain |
| Drinking water reduce headache |
| Ibuprofen is good for elbow pain |
| **Claims about aesthetic treatments** |
| Anti-cellulite cream, which is new and more expensive, makes your skin softer |
| Beer yeast to improve facial skin |
| Colgate's toothpaste makes your teeth whiter |
| **Claims about treatments for COVID-19** |
| AstraZeneca COVID-19 vaccine caused side effects (e.g., blood clots) |
| Different activities to improve health during COVID-19 confinement (e.g., sunbathing to get vitamin D, talk to neighbours to don't feel lonely, listen to music to increase motivation) |
| Hand sanitizer prevents disease |
| MMS (Miracle Mineral Solution) protects against COVID-19 |

**Fig 3. Examples of claims about treatment effects.**

- **Discrepancy in usefulness of the IHC resources for teachers**: We observed discordance in qualitative and quantitative findings about the usefulness of the IHC resources for teachers. Before starting the lessons, they expected the textbook to be useful (median score 4.5); after the lessons and at the end of the lessons, they reported that the textbook and the teachers' guide were useful (median score 5). However, during the NPOs we observed that they rarely used their printed copy of the textbook or the teachers' guide. This discrepancy could be due to the teachers rating the IHC resources as highly useful because they used them to prepare the lessons (adaptation of the IHC resources); instead, observers rated the IHC resources as less useful because teachers made many adaptations to them.

- **Complementarity in techniques used by teachers to teach the lessons**: We observed that qualitative findings complemented the quantitative findings about how teachers delivered the lessons. Teachers used the techniques of the proposed plan in the IHC resources (review of the previous lesson, reading the lesson's comic, a discussion, and the completion of the activity/exercises), and other methods or strategies. During the NPOs, we observed differences in the implementation of the techniques of the proposed plan in the IHC resources between participating schools, and we captured in detail how other strategies and methods were implemented.

### Recommendations for using the IHC resources in Barcelona primary schools

Based on the importance of the findings and their implications for changes to the IHC resources (S7.6 File in S7 File), the IHC-Barcelona Working Group agreed on the following recommendations:

- It is important that teachers plan an accurate timetable to allow enough time to complete all the lessons.

- It is important that teachers use the comic in the textbook to engage students' interest.

- Teachers can adapt the IHC resources to the teaching strategy and educational project of each school.

- The IHC resources need to include activities that promote student participation.

- It is important to collect, in an open database, examples, activities and materials developed based on the IHC resources and elaborated in different settings; in order to share teaching and learning experiences and avoid duplication of efforts.

- The availability of ICT tools in the classroom can support the development of more participatory and interactive IHC resources.

- The availability of ICT tools in the classroom can replace the use of paper based IHC resources.

## Discussion

### Summary of findings

We described in depth the experience of using IHC resources in two primary schools from Barcelona (Spain) in 2019–2020 and 2020–2021 school years. During 2019–2020 school year, COVID-19 pandemic disrupted the study; therefore, we adapted methods to the schools' COVID-19 prevention and control measures during 2020–2021 school year.

The participating schools planned and conducted the lessons using the IHC resources in different ways: one school followed the suggested IHC teaching plan and completed all the lessons (nine lessons), the other school applied an extensive project-based learning methodology and did not complete all the lessons (3–5 lessons).

Students and teachers understood, were interested and were able to apply the contents of the IHC resources. During the lessons, the textbook was useful for students; nevertheless, for the teachers, the usefulness of the IHC resources was variable. Teachers from both schools used other methods and strategies to teach the lessons, including 1) adaptation of the IHC resources (development of activities/exercises and work materials based on the IHC or on other related additional contents) to promote student participation, and 2) use of ICT tools to support teaching of the lessons.

We identified several facilitating factors mostly related to the teachers (such us the ability to adapt the style and context, self-efficacy or the understanding of the content), and some related to the students (such as attitudes and motivation). We identified few barriers related to the students (such as peer influence and attitudes) or to the school system (such as organisation and time constraints).

The quantitative and qualitative findings showed great convergence in the joint display, except for the usefulness of the IHC resources for teachers (discrepancy findings) and the techniques used by teachers to teach the lessons (complementarity findings).

We proposed seven recommendations for using the IHC resources in Barcelona primary schools, involving teachers, the IHC resources, and ICT tools.

## Our study in the context of current knowledge

Students and teachers in our setting rated positively the experience of using IHC resources to learn to think critically about health. These results are consistent with findings observed in previous studies in other settings, including one in a school in Italy [20], one in a school in Ireland [21], and two in schools in Rwanda [22, 23]. Although experiences in different settings need to be further explored, IHC resources appear to be suitable for teaching critical thinking about health to different populations of students.

Teachers in our setting adapted the IHC resources to teach the lessons to their students; however, the adaptation process showed great variability between the two participating schools (due to differences in the teaching plan, in the use of IHC resources, or in teaching techniques). This finding reflects the country's education system (decentralised model), where schools have a great deal of autonomy [46]. In this context, however, it is still necessary to assess whether it would be more efficient to develop a centralised adaptation of the IHC resources for all schools in our setting, as proposed by Glynn et al. in Ireland [21], or that each school in our setting should freely adapt the IHC resources to their needs, as happened in the present study.

The adaptation of the IHC resources mainly focused on using different teaching techniques (e.g., didactic strategies, discussion strategies, role playing, or E-learning) and developing different activities/exercises to increase student participation during lessons. A recent overview of systematic reviews identified and described 37 teaching techniques for teaching primary and secondary school students to think critically [47]. Although the certainty of the evidence for the effects of these teaching techniques is limited, interested teachers could use this evidence to identify and select the most appropriate teaching techniques to teach their students to think critically about health.

The IHC resources were adapted to be used in digital classrooms. Currently, most of the educational centres in our environment incorporate ICT as educational tools to facilitate the

teaching-learning processes [48]. The Plan for Digitalisation and Digital Competences of the Educational System (2021–2027) includes different actions aimed at digitalising educational centres and members of the educational community, such as: 1) development of the educational digital competence (students, teachers, and centres), 2) digitalisation of the educational centre, 3) creation of educational resources in digital format, and 4) advanced digital methodologies and competences [49]. In this context of digital transformation of education, it is crucial to disseminate the digital version of IHC resources (the animated version of all the lessons of the IHC primary resources are now available [50]), for example through open educational resource repositories [51].

We used a mixed methods approach to explore the students' and teachers' experience when using the IHC primary school resources. Mixed methods research combines quantitative and qualitative methods to answer a research question [30]. This approach helped us deepen our understanding of the phenomenon using different perspectives and methods [31]. Mixed methods research is an evolving methodology that is used in various disciplines, including health science and education [41, 52]; moreover, this approach could be useful to explore health promotion initiatives in schools [53, 54]. In this field, mixed methods can contribute to 1) understand the complexity of health promotion initiatives, 2) identify factors related to efficacy, 3) identify factors related to transferability from one context to another, and 4) point out challenges both from health and educational sectors [55].

Finally, circumstantially, we were able to assess the effect of the COVID-19 pandemic on some of the study outcomes. Most notably, we observed that, during the second intervention (intervention after COVID-19 pandemic lockdown), students and teachers had developed an increased ability to identify examples of claims about the effects of treatments. This finding probably reflects the impact of the COVID-19 "infodemic" (an overabundance of information, sometimes accurate and sometimes not, that makes it difficult for people to find trustworthy sources and reliable guidance when they need it [56]) in the study population [57]. This new worrying public health scenario has revealed and magnified the importance of learning to critically appraise health information and make well-informed health decisions.

## Strengths and limitations

Our study has several strengths. Firstly, before this study, a multidisciplinary group (translator, researchers, students, teachers, and medical doctors) translated the IHC resources into Spanish to fit the text of the resources to this setting. Secondly, the IHC-Barcelona Working Group gathered representatives of the different areas of interest (researchers, teachers, family members, and paediatric primary care providers) to help ensure a broad perspective on the findings and include insights from different users. Thirdly, we piloted an intervention in schools that has already been shown to be effective in a large cluster randomised trial in Uganda, where 120 schools an over 10,000 children participated [9]. Finally, we used a mixed methods approach, with quantitative and qualitative data collection and integration analysis, to obtain comprehensive and validated findings and draw more rigorous and reliable recommendations.

Our study also has some limitations. Firstly, we used a convenience sample, which is small, geographically limited, and not representative. However, we provided a transparent and detailed description of the sampling process and data collection methods to inform other stakeholders who might consider transferring our findings to different settings. Secondly, the researchers themselves conducted the qualitative data collection processes (NOPs and SSIs). To reduce the potential risk of bias, we used multiple data collection methods to explore the same phenomenon from different perspectives, and a mixed methods approach to integrate the findings. Thirdly, in this study we did not aim to assess the effect of IHC resources since

we did not include a control group or have a validated CLAIM test for our setting. Moreover, we were unable to estimate the effect with a non-validated test in our sample due to one school not completing the intervention. Finally, it was a challenge to use a mixed method approach since it requires expertise in quantitative and qualitative methods, and in the combination of both.

## Implications for practice and research

**Implications for practice.**   We formulated recommendations to support the use of the IHC resources in our setting, which involve teachers, the IHC resources, and ICT tools. Before implementing the IHC resources, each school should consider these recommendations to improve the end users' experience.

Additionally, further efforts are needed to disseminate and promote the use of the IHC resources in schools in our setting.

**Implications for research.**   It is necessary to continue exploring the barriers to use of IHC resources in our setting, especially the barriers that limit the access and participation of schools. In parallel, other strategies to teach IHC key concepts outside school contexts should be explored (e.g., summer camps).

We are currently working on other contextualization activities to ensure the relevance and appropriateness of the IHC resources for Spanish primary school children 1) systematic assessment to identify and describe relevant educational documents and resources that support teaching of critical thinking about health in Spanish primary schools [58], 2) translation of the IHC learning resources into other local language (Catalan), and 3) development and validation of an interactive test to measure the ability of Spanish primary school children to assess treatment claims and make informed health choices (the interactive CLAIM Test) [59].

## Conclusions

Students and teachers from primary schools in Barcelona showed a positive experience when using IHC resources; however, these resources should be adapted to promote classroom participation.

The COVID-19 pandemic has highlighted—more than ever—the importance of teaching and learning critical thinking about health. The IHC resources can help empower children around the world to make well-informed health decisions as adults.

## Supporting information

**S1 File. Study variables.**
(PDF)

**S2 File. Good Reporting of A Mixed Methods Study (GRAMMS) checklist.**
(PDF)

**S3 File. Workshop program.**
(PDF)

**S4 File. List of the key concepts included in the informed health choices learning resources for primary school children.**
(PDF)

**S5 File. *Ad hoc* questionnaires and guides.**
(PDF)

**S6 File. Flow chart of schools' selection process.**
(PDF)

**S7 File. Supplementary tables.**
(PDF)

**S8 File. Activities and work materials developed by teachers.** The S8 File is available at:
https://doi.org/10.6084/m9.figshare.23148563.v1.
(PDF)

## Acknowledgments

Laura Samsó Jofra is a doctoral candidate for the PhD in Methodology of Biomedical Research and Public Health, Universitat Autònoma de Barcelona, Barcelona, Spain. We would like to thank students and teachers from the two schools in Barcelona (Escola Sant Martí [http://www.escolasantmartibcn.cat], and Escola Virolai [https://www.virolai.com]) for their participation in the study. We would like to thank Dr Andrew Oxman (Centre for Epidemic Interventions Research, Norwegian Institute of Public Health, Oslo, Norway) for his advice and feedback on this manuscript.

## Author Contributions

**Conceptualization:** Laura Samsó Jofra, Sarah Rosenbaum, Laura Martínez García.

**Formal analysis:** Laura Samsó Jofra, Laura Martínez García.

**Funding acquisition:** Laura Martínez García.

**Investigation:** Laura Samsó Jofra, Laura Martínez García.

**Methodology:** Laura Samsó Jofra, Pablo Alonso-Coello, Esther Cánovas Martínez, Carol de Britos Marsal, Ana Gallego Iborra, Ena Pery Niño de Guzman Quispe, Giordano Pérez-Gaxiola, Carolina Requeijo, Marta Roqué i Figuls, Sarah Rosenbaum, Karla Salas-Gama, Iratxe Urreta-Barallobre, Laura Martínez García.

**Project administration:** Laura Martínez García.

**Supervision:** Laura Martínez García.

**Visualization:** Laura Samsó Jofra, Laura Martínez García.

**Writing – original draft:** Laura Samsó Jofra, Laura Martínez García.

**Writing – review & editing:** Laura Samsó Jofra, Pablo Alonso-Coello, Esther Cánovas Martínez, Carol de Britos Marsal, Ana Gallego Iborra, Ena Pery Niño de Guzman Quispe, Giordano Pérez-Gaxiola, Carolina Requeijo, Marta Roqué i Figuls, Sarah Rosenbaum, Karla Salas-Gama, Iratxe Urreta-Barallobre, Laura Martínez García.

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
