## [Decision Letter · Decision Letter 0]

13 Mar 2023

PONE-D-23-05069Piloting the Informed Health Choices resources in Barcelona primary schools: A mixed methods studyPLOS ONE

Dear Dr. Martínez García,

Thank you for submitting your manuscript to PLOS ONE. After careful consideration, we feel that it has merit but does not fully meet PLOS ONE’s publication criteria as it currently stands. Therefore, we invite you to submit a revised version of the manuscript that addresses the points raised during the review process.

ACADEMIC EDITOR: Please follow the instructions of the three reviewers carefully. Make substantial revisions as your paper will be thoroughly reviewed, as one reviewer recommended rejecting it at this stage.

We look forward to receiving your revised manuscript.

Kind regards,

Anastassia Zabrodskaja, Ph.D.

Academic Editor

PLOS ONE

Journal Requirements:

"This study has been funded by Instituto de Salud Carlos III [CP18/00007] (Cofunded by European Regional Development Fund/European Social Fund, 'Investing in your future'). LMG has a Miguel Servet research contract from the Institute of Health Carlos III [CP18/00007] (Cofunded by European Regional Development Fund/European Social Fund, 'Investing in your future'). Dr. Antoni Esteve Foundation has funded the Spanish translation and production of the IHC resources. The funders have not participated in either the design, or the development of the study."

"NO authors have competing interests"

Reviewers' comments:

Reviewer's Responses to Questions

**Comments to the Author**

1. Is the manuscript technically sound, and do the data support the conclusions?

Reviewer #1: Yes

Reviewer #2: Yes

Reviewer #3: Yes

2. Has the statistical analysis been performed appropriately and rigorously? 

Reviewer #1: Yes

Reviewer #2: N/A

Reviewer #3: Yes

3. Have the authors made all data underlying the findings in their manuscript fully available?

Reviewer #1: Yes

Reviewer #2: No

Reviewer #3: Yes

4. Is the manuscript presented in an intelligible fashion and written in standard English?

Reviewer #1: Yes

Reviewer #2: Yes

Reviewer #3: Yes

5. Review Comments to the Author

Reviewer #1: The paper is interesting and comprehensive in the analysis, but there are some issues worth noting.

There is no problematization of how the observations were conducted. There could have been a bias from the researchers towards seeing what they wanted to and score high. How was this ameliorated? Also, some of the teachers rank the lesson quite highly. This is could be an example of authority bias unless some alternative was provided. That said, the amount of adaptations that the teachers had to do is staggering. Were they evaluating the resources or their own versions of the resources? If the latter (as I suspect) then they were essentially asked to evaluate their own teaching, which does not answer the research questions.

PowerPoints, data projection and using Google classroom are generally not considered the height of ICT use in education. Examples of ICT use would include using augmented reality, simulations, investigations using the internet... I know some of these are not easy to implement in ordinary schools, but this is not specified in the paper, and only vague references to "interactive ICT" are present.

Reviewer #2: Peer review report on Manuscript Number: PONE-D-23-05069

Piloting the Informed Health Choices resources in Barcelona primary schools: A mixed

methods study

General

The authors have pilot tested an educational program for teaching school children critical thinking about health interventions. The program has previously been shown to be effective in a large randomised trial in Uganda, and has also been successfully piloted an few other settings. Thus, it makes sense to conduct pilot test before considering whether to implement the program in a specific setting, e.g. Barcelona/Spain.

The study is thorough, using a range of methodological approaches, together designed to give a broad picture of teachers’ and students’ experiences with the program.

The manuscript is well written, and although it’s fairly long, it is difficult to see how it could be substantially shortened. The study provide potentially useful insights for groups, both in Spain and outside, that consider implementing the same teaching program.

There are some issues that need to be dealt with before this piece is suitable for publication, but as far as I can see there are no major substantive problems – mainly issues around reporting, probably.

Specifics

Line 116:

The Escola Nova 21 intiative is mentioned, and should probably be briefly explained, if possible (though, if it is impossible to explain in one sentence, perhaps the reference that is there now, is sufficient).

Line 119:

A little surprising to learn that other schools have been part of the IHC project – the reader might expect some brief explanation.

Line 127:

I feel it natural to mention the length (in time) of the workshop here, not just under Results.

Line 131:

Seems natural to give a slightly better description of the book (i.e. cartoons, with stories designed to illustrate and explain key concepts).

Line 146:

I find it hard to understand what the ad hoc questionnaires looked like. From the description and especially the data analysis and results sections, it seems that the questionnaire data was managed quantitatively, so I assume the included some form of scoring systems, but this is something of black box.

Line 154:

I have the same issue with the ad hoc guide as for the ad hoc questionnaires (see previous point).

Line 185:

I think you should specify that you only considered the CLAIM test results for the students AT SCHOOLS WHERE THEY completed all the lessons.

Line 272:

I don’t find it natural to compare the scores across the different domains, so I suggest re-writing “they thought that students would be slightly less interested in them”, e.g. “while the teachers’ score for desirability among the students was 3.5” (you can probably phrase this better than me).

Line 341:

The results-section on barriers and facilitators is too difficult to understand, and this is probably related to my earlier comments on not understanding the ad hoc questionnaire/guide. Although an explanation is given under the table, the last column is not comprehendible to me. I assume the “n” and the percentage are quantitative results of some kind from the guide and questionnaire, but they don’t mean anything to me unless there is a slightly more comprehensive explanation.

Line 482:

This bit of text is interesting, and a little bit confusing:

“We are currently working on other contextualization projects to complement the findings of this study. We are translating the IHC primary school resources into Catalan (one of the official local languages in Spain), conducting a context analysis [28], and adapting and validating a CLAIM Test. This research will allow us to ensure the relevance and appropriateness of the IHC resources for the Spanish education system.

Conclusions

It is feasible to use the IHC resources in Barcelona primary schools; however, these resources should be adapted to promote classroom participation.”

So, does this mean that the authors view the current study as supportive of the suitability of the IHC-program for schools in Barcelona (first sentence in the Conclusion), but perhaps less clearly so for Spanish schools more widely? This seems to be the message, though it’s not stated explicitly. However, if this is the case it makes little sense to prioritise translation into Catalan in order to “allow us to ensure the relevance and appropriateness of the IHC resources for the Spanish education system.” Probably not a major problem, but there seems to be a slight lack of logic in this particular phase.

Minor detail: The Centre for Informed Health Choices does not exist any longer, but has become part of the larger Centre for Epidemic Interventions Research (this should be changed, a couple of places in the manuscript).

Final thing: The link to Supporting Information 10S6-fil (page 49) is very slow to respond (many minutes).

Reviewer #3: I thank the authors for submitting this manuscript which touches on a very important topic. There are many good things to say about the manuscript, for example, it is overall well-written, clear, and concise for the most part. Overall, the objectives, methods, and results are well-described. I will concentrate on a few issues that I think if addressed would greatly improve the understanding of its methods, findings, and recommendations, especially by a reader who might be hearing about this work for the first time. The issues I have highlighted below should be interpreted simply as areas of improvement for an article that is overall well-written and for which the authors should be congratulated.

The aim of this study is stated as “to explore students’ and teachers’ experiences when using the IHC resources in primary schools in Barcelona” and to formulate recommendations to use the IHC primary school resources in this setting. However, the conclusion is that “it is feasible.” Was feasibility part of the purpose of the study? If so, it should be clearly stated. Additionally, the methods and results should be written in a manner that speaks to feasibility if this was an aim.

Background: More information should be provided to motivate the study. Even if these might already be enumerated elsewhere, a statement alluding to this would greatly help a reader unfamiliar with the topic to understand the nature of the problem you would like to address, and why this study is essential to addressing the problem.

The authors state: “We used the Standards for Reporting Qualitative Research to integrate and report the quantitative and qualitative findings as there is not yet a formal checklist for reporting mixed methods studies.” This statement might not be accurate. Guidance does exist on how to plan for and report mixed methods studies.

Another tool, among many others that could have been used, is the Consolidated criteria for reporting qualitative research (COREQ). It would be great if the authors helped a reader to understand their thinking in selecting the tool (SRQR) they used for reporting. With this said, there are checklists and tools that can be used to report mixed methods studies. For example, did the authors consider the Good Reporting of A Mixed Methods Study (GRAMMS) or the Mixed Methods Article Reporting Standards (MMARS) checklists at all? I understand the need to report qualitative results, however, this was a mixed methods study. Therefore, one would expect that a more fitting tool to the study design would be used. Using a heavily quantitative tool to report a mixed methods study could result in reporting that is skewed against one method. The purpose of a mixed methods study is to report findings from each method employed individually as well as both methods combined.

There is agreement that mixed methods studies should highlight the interconnection between the two sources of data and should demonstrate the collective advantage of their combined use provides a better understanding of the research problem than a single source.”

To this end, I would have expected the authors to:

• Provide a brief justification for using mixed methods for this research question;

• describe each method in terms of sampling, participants, data collection, and analysis;

• Highlight where the integration of the methods was done, how it occurred, what was found, and how using a second method helped to explain the findings. One would want to know if using a second method was better than using one. What insights if any, were gained from mixing the methods?

• Describe any limitation, if any, of using both methods, to the overall project or any limitation to one method that might have resulted from integrating it with a second method, if any. This basically asks the question: is using two methods always advantageous? It would be great if the researchers reflected on this and provided a brief statement about their experiences/observations implementing a mixed methods design for this study.

Criteria for selecting schools are listed: “We used the following eligibility criteria: 1)

schools included in the school directory from the Regional Ministry of Education from the Government ofCatalonia (2018-2019) [33]; 2) schools that had previously participated in a health promotion programme (2016-2017) [34]; and 3) schools that had previously participated in the initiative Escola Nova 21 [35]. We also took into consideration whether the schools included students that were representative of the neighbourhood, if they were in different neighbourhoods of the city, their type of funding (public, publicly funded private, or private schools), and schools that previously participated in the IHC project.” One might wonder why schools that participated in these programmes were selected. What was the relevance of the programs to the IHC work? Is it likely that the researchers might have obtained different results if they had chosen schools based on a different set of criteria? I think it would be important to demonstrate to the reader why it was necessary to use these programs as part of the selection criteria. For example, one could say, the schools selected already had infrastructure from the previous projects that was relevant to IHC work. It would be greatly appreciated if the researchers highlighted the reasons and provided some context about the selection of the schools.

Line 126. Workshop with the teachers: Similar to how the authors describe the IHC resources, it would be more informative if the contents of this workshop were described in more detail. It does not have to be a long description, but a sentence or two that enable a reader to understand what was done would be helpful.

Methods for qualitative analysis: Overall, this section is fairly well-described. However, I have three main comments:

1. On line 190, the authors state that they used inductive thematic analysis, yet they go on to describe that they “identified themes related to understandability, desirability, suitability, and usefulness of the IHC resources for students and teachers; the technique used to teach the lessons; facilitators and barriers to teaching the lessons; and suggestions to improve the lessons.” This does not seem like inductive thematic analysis. Rather it seems to be a framework analysis, as the themes seem to have been pre-determined is some sense. However, assuming that this was a framework analysis, it is still not well-described. Inductive If in doubt, refer to the article “Demonstrating Rigor Using Thematic Analysis: A Hybrid Approach of Inductive and Deductive Coding and Theme Development” for a description of how this could be done. Additionally, there are numerous resources that the authors could use to help clarify what analysis method they employed. Currently, this part of the article creates some confusion as to the methods used. Relatedly, the authors seem to have used the user-experiences framework. If they did, they should state so and provide the relevant citation.

2. It would be good to state how any disagreements in coding or interpretation were addressed.

3. The user experience framework used, in my opinion, should be explained a bit more.

The main objective of the study is stated as assessing experience using the IHC resources. Was the CLAIM evaluation tool also used to assess user experience? It appears, from the methods and the way the results are reported that there were perhaps other objectives. If this is so, it should be stated, and objectives clarified. If not, one might wonder how some of the methods employed are related to the study objectives.

Quantitative analysis: “We conducted a descriptive analysis of the categorical variables (absolute and relative frequencies), and the continuous variables (median and range)” Which categorical variables did you apply these methods to? For example, was it applied to demographic categorical variables or all?

Table 1. Do the authors have information about the age and gender of participants and any other relevant demographic information? If so, it would be helpful to include it in the table.

Line 271-272: “Before starting the lessons, teachers expected that students would understand and would be able to apply the content of the lessons in their daily lives (median score 4 in these items [1 meaning completely disagree and 5 meaning completely agree]),” This statement seems to imply that the authors used some kind of scale to score the responses. It is not clear which scale was used. Was it a Likert-type scale? If so, this should be clearly described in the methods section and its analysis plan should be clearly described. The quantitative results seem to come out of the blue. They do not seem to have a systematic methodology. If they do, it was not clearly described to enable a reader to understand. In line with my comments about reflecting on the value added by using both qualitative and quantitative methods, I suggest that the authors provide more information about the quantitative methods of data collection and analysis. This is partly what I alluded to in my comments about skewed reporting when a mainly qualitative checklist I used to plan for a mixed methods study. One methodology might suffer, as seems to be the case here.

Discussion: Please include a section on reflexivity and discuss the interrelatedness of the quantitative and qualitative findings.

6. PLOS authors have the option to publish the peer review history of their article (what does this mean?). If published, this will include your full peer review and any attached files.

Reviewer #1: **Yes: **Elena Prieto-Rodriguez

Reviewer #2: **Yes: **Atle Fretheim

Reviewer #3: No

---

## [Author Response · Author response to Decision Letter 0]

25 May 2023

Comments from the Journal Editor

Comment 1. PLOS ONE's style requirements

Response 1

We have reviewed and modified the manuscript’s style and the file’s name to meet the style requirements of PLOS ONE.

Comment 2. Financial Disclosure Statement

Thank you for stating the following financial disclosure: "This study has been funded by Instituto de Salud Carlos III [CP18/00007] (Cofunded by European Regional Development Fund/European Social Fund, 'Investing in your future'). LMG has a Miguel Servet research contract from the Institute of Health Carlos III [CP18/00007] (Cofunded by European Regional Development Fund/European Social Fund, 'Investing in your future'). Dr. Antoni Esteve Foundation has funded the Spanish translation and production of the IHC resources. The funders have not participated in either the design, or the development of the study."

Response 2

We have reviewed and modified the text of “Financial Disclosure Statement” section according to the journal editor’s suggestion.

The text now reads: “This study has been funded by Instituto de Salud Carlos III [CP18/00007] (Cofunded by European Regional Development Fund/European Social Fund, 'Investing in your future'). LMG has a Miguel Servet research contract from the Institute of Health Carlos III [CP18/00007] (Cofunded by European Regional Development Fund/European Social Fund, 'Investing in your future'). Dr. Antoni Esteve Foundation has funded the Spanish translation and production of the IHC resources. The funders had no role in study design, data collection and analysis, decision to publish, or preparation of the manuscript.”

Additionally, we have included this statement in the cover latter.

Comment 3. Competing Interests

Thank you for stating the following in your Competing Interests section: "NO authors have competing interests"

Response 3

We have reviewed and modified the text of “Competing interests” section according to the journal editor’s suggestion.

The text now reads: “The authors have declared that no competing interests exist.”

Additionally, we have included this statement in the cover latter.

Comment 4. Ethics statement

Your ethics statement should only appear in the Methods section of your manuscript. If your ethics statement is written in any section besides the Methods, please delete it from any other section.

Response 4

We have reviewed the text of “Ethics statement” according to the journal editor’s suggestion. Now the ethical statement is only included in “Methods” section.

Comment 5. Figures

Please include a separate caption for each figure in your manuscript.

Response 5

We have included the figures in the manuscript according to the journal editor’s suggestion.

Comments from Reviewer 1

Comment 6. Risk of bias

There is no problematization of how the observations were conducted. There could have been a bias from the researchers towards seeing what they wanted to and score high. How was this ameliorated?

Response 6

We already described the assessment and management of the potential risk of bias in “Discussion; Strengths and limitations” section. Nevertheless, we have reviewed and modified the text to better address the potential risk of bias.

The text now reads: “Secondly, the researchers themselves conducted the qualitative data collection processes (NOPs and SSIs). To reduce the potential risk of bias, we used multiple data collection methods to explore the same phenomenon from different perspectives, and a mixed methods approach to integrate the findings.”

Comment 7. Adaptation of the IHC resources

Also, some of the teachers rank the lesson quite highly. This could be an example of authority bias unless some alternative was provided. That said, the amount of adaptations that the teachers had to do is staggering. Were they evaluating the resources or their own versions of the resources? If the latter (as I suspect) then they were essentially asked to evaluate their own teaching, which does not answer the research questions.

Response 7

The main objective of the study is to explore the students’ and teachers’ experience when using the IHC primary school resources. Thus, to evaluate the resources themselves or the teaching with the resources was outside the scope of the study.

In fact, the adaptation of the resources was an important finding of the assessment of teachers’ experience with the IHC resources in this context. For this reason, the following recommendation was included in the "Results; Recommendations for using the IHC resources in Barcelona primary schools” section: “Teachers can adapt the IHC resources to the teaching strategy and educational project of each school.”

Comment 8. Information and Communications Technologies

PowerPoints, data projection and using Google classroom are generally not considered the height of ICT use in education. Examples of ICT use would include using augmented reality, simulations, investigations using the internet... I know some of these are not easy to implement in ordinary schools, but this is not specified in the paper, and only vague references to "interactive ICT" are present.

Response 8

We have replaced the term “Information and Communications Technologies (ICT)” for “Information and Communications Technologies (ICT) tools (including equipment [hardware] and computer programs [software])” to improve the clarity of the text.

Comments from Reviewer 2

Comment 9. Escola Nova 21 initiative

Line 116: The Escola Nova 21 initiative is mentioned, and should probably be briefly explained, if possible (though, if it is impossible to explain in one sentence, perhaps the reference that is there now, is sufficient).

Response 9

We have added a brief description of the initiative Escola Nova 21 in “Methods; Setting; Selection of schools” section according to the reviewer’s suggestion.

The text now reads: “We selected a convenience sample of three schools in Barcelona. We used the following eligibility criteria: 1) schools included in the school directory from the Regional Ministry of Education from the Government of Catalonia (2018-2019) [35]; 2) schools that had previously participated in a health promotion programme (2016-2017) [36]; and 3) schools that had previously participated in the initiative Escola Nova 21 (alliance of schools and civil society institutions for updating the Catalan education system during 2016-2019) [37].”

Comment 10. Schools that previously participated in the IHC project

Line 119: A little surprising to learn that other schools have been part of the IHC project – the reader might expect some brief explanation.

Response 10

We have reviewed and modified the text related to the school’s participation in the IHC project in “Methods; Setting; Selection of schools” section to improve the clarity of the text.

The text now reads: “We also took into consideration whether the schools included students that were representative of the neighbourhood, if they were in different neighbourhoods of the city, their type of funding (public, publicly-funded private, or private schools), and schools that previously participated in the IHC resources translation into Spanish.”

Comment 11. Workshop

Line 127: I feel it natural to mention the length (in time) of the workshop here, not just under Results.

Response 11

We have added a brief description of the workshop in “Methods; Setting; Intervention in the schools” section according to the reviewer’s suggestion.

The text now reads: “The workshop program included the following sessions: 1) an opening session about Evidence-Based Medicine, 2) the presentation of the IHC project and the IHC resources, 3) the presentation of the study, 4) a mock lesson to the teachers, and 5) teachers’ presentation about their plan to teach the lessons to the students (the workshop program is available in S3 File). The workshop lasted five hours.”

Comment 12. Textbook

Line 131: Seems natural to give a slightly better description of the book (i.e. cartoons, with stories designed to illustrate and explain key concepts).

Response 12

We have added a brief description of the textbook in “Methods; Setting; Intervention in the schools” section according to the reviewer’s suggestion.

The text now reads: “The textbook tells a story, narrated as a comic, about a brother and a sister, John and Julie, who meet two teachers and health researchers, professor Compare and professor Fair. The professors teach the children: 1) what questions they should ask when someone says something about a treatment; 2) what questions health researchers ask to find out more about treatment effects; and 3) what questions they should ask when deciding to use a treatment or not.”

Additionally, we have added the following figure and supporting information file:

• Figure “Fig 1. The Informed Health Choices learning resources for primary school children (English and Spanish version) [10-12, 27-29]”

• Supporting information file “S4 File. List of the key concepts included in the Informed Health Choices learning resources for primary school children”

Comment 13. Ad hoc questionnaires and guide

Line 146: I find it hard to understand what the ad hoc questionnaires looked like. From the description and especially the data analysis and results sections, it seems that the questionnaire data was managed quantitatively, so I assume the included some form of scoring systems, but this is something of black box.

Line 154: I have the same issue with the ad hoc guide as for the ad hoc questionnaires (see previous point).

Response 13

We have added the questionnaires and guides in “Supporting information; S5 File. Ad hoc questionnaires and guides” section to improve the clarity of the manuscript.

The text now reads: 

• “S5.1 File. Questionnaire for the assessment of the IHC resources by the teachers before the lessons

o S5.2 File. Questionnaire for the assessment of the lessons by the teachers after a lesson

o S5.3 File. Questionnaire for the overall assessment of the IHC resources by the teachers at the end of the lessons

o S5.4 File. Guide for the non-participatory observations during the lessons

o S5.5 File. Guide for the semi-structured interviews with the students after a lesson”

Comment 14. CLAIM test analysis

Line 185: I think you should specify that you only considered the CLAIM test results for the students AT SCHOOLS WHERE THEY completed all the lessons.

Response 14

We have reviewed and modified the explanation of the CLAIM test analysis in “Methods; Data analysis; Quantitative analysis” section according to the reviewer’s suggestion.

The text now reads: “We only considered the CLAIM test results for the students at schools that completed all the lessons.”

Comment 15. Scores across domains

Line 272: I don’t find it natural to compare the scores across the different domains, so I suggest re-writing “they thought that students would be slightly less interested in them”, e.g. “while the teachers’ score for desirability among the students was 3.5” (you can probably phrase this better than me).

Response 15

We have reviewed and modified the explanation of the scores across domains in “Results; Students’ and teachers’ experience with the IHC resources; Quantitative findings” section according to the reviewer’s suggestion.

The text now reads: “Before starting the lessons, teachers expected that students would understand and would be able to apply the content of the lessons in their daily lives (median score 4 [1 meaning completely disagree and 5 meaning completely agree] in assessment before the lessons for both items), but they thought that students would be slightly less interested in them (median score 3.5 in assessment before the lessons) (Table 2). During the lessons, students understood (median score 5 in assessment after a lesson), were interested (median score 4 in assessment after a lesson), and were able to apply the content of the lessons (median score 4 in assessment after a lesson). Teachers understood the content of the IHC resources (median score 5 in all the assessments), found them interesting (median score 4.5 in assessment before the lessons, and median score 5 in assessment after a lesson and at the end of the lessons) and suitable (median score 4 in assessment before the lessons, 5 in assessment after a lesson, and 4.5 at the end of the lessons).

In terms of usefulness of the IHC resources, teachers thought that the IHC resources would be useful for the students (median score 4 in assessment before the lessons, 5 for the textbook in assessment after a lesson, and 5 at the end of the lessons). Before starting the lessons, the teachers mainly expected the textbook to be useful (median score 4.5 in assessment before the lessons); after the lessons and at the end of the lessons, they reported that both, the textbook and the teachers’ guide, were useful (median score 5 for the textbook and the teachers’ guide in both assessments). Teachers found the poster not useful, neither for the students nor for themselves.”

Comment 16. Facilitators and barriers

Line 341: The results-section on barriers and facilitators is too difficult to understand, and this is probably related to my earlier comments on not understanding the ad hoc questionnaire/guide. Although an explanation is given under the table, the last column is not comprehendible to me. I assume the “n” and the percentage are quantitative results of some kind from the guide and questionnaire, but they don’t mean anything to me unless there is a slightly more comprehensive explanation.

Response 16

We have reviewed and modified the explanation of the facilitators and barriers in the footnote of Table 3 according to the reviewer’s suggestion. Consequently, we have also corrected the absolute and relative frequencies according to the new explanation.

The text in footnote of Table 3 now reads: “n (%): times the facilitator/barrier has been reported by teachers in a total of 26 assessments after lessons.”

Comment 17. Contextualization projects

Line 482: This bit of text is interesting, and a little bit confusing: “We are currently working on other contextualization projects to complement the findings of this study. We are translating the IHC primary school resources into Catalan (one of the official local languages in Spain), conducting a context analysis [28], and adapting and validating a CLAIM Test. This research will allow us to ensure the relevance and appropriateness of the IHC resources for the Spanish education system.”

Response 17

We have reviewed and modified the explanation of the contextualization of the Informed Health Choices project in “Background; Contextualization of the Informed Health Choices resources” and “Discussion; Implications for practice and research; Implications for research” sections according to the reviewer’s suggestion.

In the following sections, the text now reads:

• “Background; Contextualization of the Informed Health Choices resources” section: “The contextualization of the IHC involves activities to explore how these resources can be used in a different setting from the one that they were originally designed for (primary schools in Uganda) [14]. The IHC Working Group proposed several contextualization activities 1) context analysis to explore educational conditions in primary schools to teach critical thinking about health, 2) translation of the IHC resources into local language, 3) pilot testing of the IHC resources in primary schools to explore user experience, 4) adaptation of the IHC resources to improve users’ experience (if needed), and 5) translation and validation of the CLAIM Evaluation Tools to measure the ability of primary school children to assess treatment claims and make informed health choices [14].

Different working groups around the world are contextualizing the IHC resources into their settings [15-19]. So far, the main contextualization activities performed include 1) translation of the IHC resources (available on the IHC website in 13 languages) [5] [IHC website 2022], 2) pilot testing of the IHC resources in primary schools [20-23], and 3) translation and validation of the CLAIM Evaluation Tools [24-26].

Currently, there are no specific learning resources to teach primary school children to think critically about their health in Spanish context. The IHC-Barcelona Working Group translated the IHC resources into Spanish [27-29]. The next step was to pilot test these resources and ensure their appropriateness for Spanish primary school children. For this purpose, we conducted this mixed methods study aimed to 1) explore the students’ and teachers’ experience when using the IHC primary school resources in Barcelona (Spain), and 2) formulate recommendations to use the IHC primary school resources in this setting.”

• “Discussion; Implications for practice and research; Implications for research” section: “We are currently working on other contextualization activities to ensure the relevance and appropriateness of the IHC resources for Spanish primary school children 1) systematic assessment to identify and describe relevant educational documents and resources that support teaching of critical thinking about health in Spanish primary schools [58], 2) translation of the IHC learning resources into other local language (Catalan), and 3) development and validation of an interactive test to measure the ability of Spanish primary school children to assess treatment claims and make informed health choices (the interactive CLAIM Test) [59].”

Comment 18. Feasibility

Conclusions: It is feasible to use the IHC resources in Barcelona primary schools; however, these resources should be adapted to promote classroom participation.”

So, does this mean that the authors view the current study as supportive of the suitability of the IHC-program for schools in Barcelona (first sentence in the Conclusion), but perhaps less clearly so for Spanish schools more widely? This seems to be the message, though it’s not stated explicitly. However, if this is the case it makes little sense to prioritise translation into Catalan in order to “allow us to ensure the relevance and appropriateness of the IHC resources for the Spanish education system.” Probably not a major problem, but there seems to be a slight lack of logic in this particular phase.

Response 18

We have reviewed and modified the conclusions in “Abstract; Conclusions” and “Discussion; Conclusions” sections to improve the clarity of the text.

The text now reads: “Students and teachers from primary schools in Barcelona showed a positive experience when using IHC resources; however, these resources should be adapted to promote classroom participation.”

Comment 19. Centre for Epidemic Interventions Research

The Centre for Informed Health Choices does not exist any longer, but has become part of the larger Centre for Epidemic Interventions Research (this should be changed, a couple of places in the manuscript).

Response 19

We have replaced the term “Centre for Informed Health Choices” for “Centre for Epidemic Interventions Research” according to the reviewer’s suggestion.

Comment 20 Supporting Information link

The link to Supporting Information 10S6-fil (page 49) is very slow to respond (many minutes).

Response 20

We have included the “S8 File. Activities and work materials developed by teachers” in a data repository: (https://doi.org/10.6084/m9.figshare.23148563.v1).

Comments from Reviewer 3

Comment 21. Feasibility

The aim of this study is stated as “to explore students’ and teachers’ experiences when using the IHC resources in primary schools in Barcelona” and to formulate recommendations to use the IHC primary school resources in this setting. However, the conclusion is that “it is feasible.” Was feasibility part of the purpose of the study? If so, it should be clearly stated. Additionally, the methods and results should be written in a manner that speaks to feasibility if this was an aim.

Response 21

Please see the responses to “Comment 18”.

Comment 22. Background

Background: More information should be provided to motivate the study. Even if these might already be enumerated elsewhere, a statement alluding to this would greatly help a reader unfamiliar with the topic to understand the nature of the problem you would like to address, and why this study is essential to addressing the problem.

Response 22

Please see the responses to “Comment 17”, specifically modifications in “Background; Contextualization of the Informed Health Choices resources” section.

Comment 23. Reporting checklist

The authors state: “We used the Standards for Reporting Qualitative Research to integrate and report the quantitative and qualitative findings as there is not yet a formal checklist for reporting mixed methods studies.” This statement might not be accurate. Guidance does exist on how to plan for and report mixed methods studies.

Another tool, among many others that could have been used, is the Consolidated criteria for reporting qualitative research (COREQ). It would be great if the authors helped a reader to understand their thinking in selecting the tool (SRQR) they used for reporting. With this said, there are checklists and tools that can be used to report mixed methods studies. For example, did the authors consider the Good Reporting of A Mixed Methods Study (GRAMMS) or the Mixed Methods Article Reporting Standards (MMARS) checklists at all? I understand the need to report qualitative results, however, this was a mixed methods study. Therefore, one would expect that a more fitting tool to the study design would be used. Using a heavily quantitative tool to report a mixed methods study could result in reporting that is skewed against one method. The purpose of a mixed methods study is to report findings from each method employed individually as well as both methods combined.

Response 23

Thank you for your comment and for the suggested references. After checking all the reviewers' comments and modifying the manuscript accordingly, we have used the Good Reporting of A Mixed Methods Study (GRAMMS) checklist to report the new version of the manuscript. The GRAMMS checklist is available in S2 File.

We also modified the text in “Methods; Design” section; the text now reads: “We used the Good Reporting of A Mixed Methods Study (GRAMMS) checklist to report the paper (S2 File) [33].”

Comment 24. Mixed methods studies

There is agreement that mixed methods studies should highlight the interconnection between the two sources of data and should demonstrate the collective advantage of their combined use provides a better understanding of the research problem than a single source.

To this end, I would have expected the authors to:

• Comment 24A. Justification for using mixed methods

Provide a brief justification for using mixed methods for this research question.

• Comment 24B. Methods for quantitative and qualitative approach

Describe each method in terms of sampling, participants, data collection, and analysis.

• Comment 24C. Mixed methods integration analysis and findings

Highlight where the integration of the methods was done, how it occurred, what was found, and how using a second method helped to explain the findings. One would want to know if using a second method was better than using one. What insights if any, were gained from mixing the methods?

• Comment 24D. Limitations

Describe any limitation, if any, of using both methods, to the overall project or any limitation to one method that might have resulted from integrating it with a second method, if any. This basically asks the question: is using two methods always advantageous? It would be great if the researchers reflected on this and provided a brief statement about their experiences/observations implementing a mixed methods design for this study.

Response 24A

We have reviewed and modified the justification for using a mixed methods approach in “Methods; Design” section according to the reviewer’s suggestion.

The text now reads: “We conducted a convergent mixed methods study [30]. We used multiple approaches to collect in parallel quantitative data (teachers’ questionnaires) and qualitative data (lessons’ non-participatory observations [NPOs], and students’ semi-structured interviews [SSIs]) on users’ experience with the IHC resources (S1 File). The quantitative and qualitative data were collected, analysed, and interpreted separately. Finally, we integrated the qualitative and quantitative findings for an in-depth understanding of the same phenomenon (triangulation purpose) [31].”

Response 24B

We have reviewed and modified the description of quantitative and qualitative methods in “Methods” section according to the reviewer’s suggestion. Now, each section (participants, data collection, and data analysis) includes a quantitative and qualitative subsection.

Please, see how the text now reads in the new version of the manuscript (we do not include the text in this document due to its length).

Response 24C

We have added the sections “Methods; Data analysis; Mixed methods integration analysis” and “Results; Students’ and teachers’ experience with the IHC resources; Mixed methods integration findings” sections according to the reviewer’s suggestion.

In the following sections, the text reads:

• “Methods; Data analysis; Mixed methods integration analysis” section: “We merged quantitative and qualitative results using a joint display to compare and validate the findings [41]. We applied the following steps: 1) mapping quantitative and qualitative results by outcome into a summarized matrix, 2) exploring the convergence (findings from quantitative and qualitative approach agree), complementarity (findings from each approach offer complementary information), or discrepancy (findings from each approach appear to be contradictive) [42], and 3) narrative synthesising of the integration findings. One reviewer conducted the integration analysis independently. A second reviewer cross-checked the summarised results matrix and the narrative synthesis. Disagreements were initially solved by consensus; if necessary, a third reviewer was consulted.”

• “Results; Students’ and teachers’ experience with the IHC resources; Mixed methods integration findings” section: “When integrating the qualitative and quantitative findings into a summarized matrix, we observed almost a complete convergence of the results (S7.5 File). Although, two integration concepts emerged:

o Discrepancy in usefulness of the IHC resources for teachers: We observed discordance in qualitative and quantitative findings about the usefulness of the IHC resources for teachers. Before starting the lessons, they expected the textbook to be useful (median score 4.5); after the lessons and at the end of the lessons, they reported that the textbook and the teachers’ guide were useful (median score 5). However, during the NPOs we observed that they rarely used their printed copy of the textbook or the teachers’ guide. This discrepancy could be due to the teachers rating the IHC resources as highly useful because they used them to prepare the lessons (adaptation of the IHC resources); instead, observers rated the IHC resources as less useful because teachers made many adaptations to them.

o Complementarity in techniques used by teachers to teach the lessons: We observed that qualitative findings complemented the quantitative findings about how teachers delivered the lessons. Teachers used the techniques of the proposed plan in the IHC resources (review of the previous lesson, reading the lesson’s comic, a discussion, and the completion of the activity/exercises), and other methods or strategies. During the NPOs, we observed differences in the implementation of the techniques of the proposed plan in the IHC resources between participating schools, and we captured in detail how other strategies and methods were implemented.”

Response 24D

We have added the limitations of using a mixed method approach in “Discussion; Strengths and limitations” section according to the reviewer’s suggestion.

The text reads: “Finally, it was a challenge to use a mixed method approach since it requires expertise in quantitative and qualitative methods, and in the combination of both.”

Comment 25. Selection of schools

Criteria for selecting schools are listed: “We used the following eligibility criteria: 1) schools included in the school directory from the Regional Ministry of Education from the Government of Catalonia (2018-2019) [33]; 2) schools that had previously participated in a health promotion programme (2016-2017) [34]; and 3) schools that had previously participated in the initiative Escola Nova 21 [35]. We also took into consideration whether the schools included students that were representative of the neighbourhood, if they were in different neighbourhoods of the city, their type of funding (public, publicly funded private, or private schools), and schools that previously participated in the IHC project.” 

One might wonder why schools that participated in these programmes were selected. What was the relevance of the programs to the IHC work? Is it likely that the researchers might have obtained different results if they had chosen schools based on a different set of criteria? I think it would be important to demonstrate to the reader why it was necessary to use these programs as part of the selection criteria. For example, one could say, the schools selected already had infrastructure from the previous projects that was relevant to IHC work. It would be greatly appreciated if the researchers highlighted the reasons and provided some context about the selection of the schools.

Response 25

We have reviewed the description of the of the schools’ selection process in “Methods; Setting; Selection of schools” section according to the reviewer’s suggestion.

We have included the following text: “Some of these criteria were established to identify schools that had previously collaborated on related initiatives (health promotion programmes, initiative Escola Nova 21, or the IHC project), and thus guarantee participation in the study and intervention adherence.”

Comment 26. Workshop

Line 126. Workshop with the teachers: Similar to how the authors describe the IHC resources, it would be more informative if the contents of this workshop were described in more detail. It does not have to be a long description, but a sentence or two that enable a reader to understand what was done would be helpful.

Response 26

Please see the responses to “Comment 11”.

Comment 27. Qualitative analysis

Methods for qualitative analysis: Overall, this section is fairly well-described. However, I have three main comments:

• On line 190, the authors state that they used inductive thematic analysis, yet they go on to describe that they “identified themes related to understandability, desirability, suitability, and usefulness of the IHC resources for students and teachers; the technique used to teach the lessons; facilitators and barriers to teaching the lessons; and suggestions to improve the lessons.” This does not seem like inductive thematic analysis. Rather it seems to be a framework analysis, as the themes seem to have been pre-determined is some sense. However, assuming that this was a framework analysis, it is still not well-described. Inductive If in doubt, refer to the article “Demonstrating Rigor Using Thematic Analysis: A Hybrid Approach of Inductive and Deductive Coding and Theme Development” for a description of how this could be done. Additionally, there are numerous resources that the authors could use to help clarify what analysis method they employed. Currently, this part of the article creates some confusion as to the methods used. Relatedly, the authors seem to have used the user-experiences framework. If they did, they should state so and provide the relevant citation.

• It would be good to state how any disagreements in coding or interpretation were addressed.

• The user experience framework used, in my opinion, should be explained a bit more.

Response 27

We have reviewed and modified the text related to qualitative analysis in “Methods; Data analysis; Qualitative analysis” according to the reviewer’s suggestion.

The text now reads: “We analysed qualitative data derived from lessons’ NPOs, students’ SSIs, and free-text responses of teachers’ questionnaires (S1 File). We conducted a framework deductive analysis [40] for qualitative data related to several domains of an adapted version of a user-experience honeycomb framework: understandability, desirability, suitability, and usefulness [38]. We applied the following steps: 1) categorisation of quotes using the framework’s themes, and 2) proposal of subthemes under the themes. We conducted a thematic inductive analysis for qualitative data not suitable for the framework: technique used to teach the lessons, facilitators and barriers to teach the lessons, examples of claims about treatment effects, and suggestions to improve the lessons. We applied the following steps: 1) codification of quotes, 2) proposal of descriptive themes, and 3) identification of analytic themes. One researcher categorised and coded quotes, and proposed themes and subthemes independently. A second researcher cross-checked codes, corresponding quotes, themes, and subthemes. Disagreements were initially solved by consensus; if necessary, a third reviewer was consulted.”

Comment 28. CLAIM Test

The main objective of the study is stated as assessing experience using the IHC resources. Was the CLAIM evaluation tool also used to assess user experience? It appears, from the methods and the way the results are reported that there were perhaps other objectives. If this is so, it should be stated, and objectives clarified. If not, one might wonder how some of the methods employed are related to the study objectives.

Response 28

We have reviewed the text related to CLAIM Test according to the reviewer’s suggestion.

After a thorough examination, we have considered that the assessment of CLAIMs about treatments by the students at the end of the lessons (CLAIM test) was part of the intervention itself. For this reason, we have included the text related to CLAIM Test in “Methods; Setting; Intervention in the schools” and in “Results; Intervention in the schools” sections.

Comment 29. Categorical variables

Quantitative analysis: “We conducted a descriptive analysis of the categorical variables (absolute and relative frequencies), and the continuous variables (median and range)” Which categorical variables did you apply these methods to? For example, was it applied to demographic categorical variables or all?

Response 29

We have added the type of variables in “Supporting information; S1 File. Study variables” section according to the reviewer’s suggestion.

Please, see how the table is now presented in the new version of the manuscript (we do not include the table in this document due to its length).

Comment 30. Demographic information

Table 1. Do the authors have information about the age and gender of participants and any other relevant demographic information? If so, it would be helpful to include it in the table.

Response 30

We have added the available demographic information in the “Results” section according to the reviewer’s suggestion.

In the following sections, the text now reads:

• “Results; Intervention in the schools; Participants” section:

o “The IHC-Barcelona Working Group included 21 participants (researchers, teachers, family members, and paediatric primary care providers) (Table 1), most of them were women (18/21; 85.7%).”

o “A higher proportion of students were 10 to 11-year-olds (100/143; 69.9%), and there were more boys (81/143; 56,6%) than girls.”

o “Six teachers participated in the study; all of them were women.”

• “Results; Qualitative findings” section: “A higher proportion of interviewed students were 10 to 11-year-olds (9/14; 64.3%), and the majority were girls (8/14; 57.61%).”

Comment 31. 5-Point Scale

Line 271-272: “Before starting the lessons, teachers expected that students would understand and would be able to apply the content of the lessons in their daily lives (median score 4 in these items [1 meaning completely disagree and 5 meaning completely agree]),” 

This statement seems to imply that the authors used some kind of scale to score the responses. It is not clear which scale was used. Was it a Likert-type scale? If so, this should be clearly described in the methods section and its analysis plan should be clearly described. The quantitative results seem to come out of the blue. They do not seem to have a systematic methodology. If they do, it was not clearly described to enable a reader to understand. In line with my comments about reflecting on the value added by using both qualitative and quantitative methods, I suggest that the authors provide more information about the quantitative methods of data collection and analysis. This is partly what I alluded to in my comments about skewed reporting when a mainly qualitative checklist I used to plan for a mixed methods study. One methodology might suffer, as seems to be the case here.

Response 31

We have added a brief description of the 5-Point Scale in “Methods; Data collection; Quantitative data” section according to the reviewer’s suggestion.

The text reads: “In this assessment, and in the following ones, the understandability, desirability, suitability, and usefulness of the IHC resources were evaluated using a 5-Point Scale (1 meaning completely disagree and 5 meaning completely agree).”

Comment 32. Mixed methods integration findings in discussion

Discussion: Please include a section on reflexivity and discuss the interrelatedness of the quantitative and qualitative findings.

Response 32

We have added a reflexion related to mixed methods approach in “Discussion; Our study in the context of current knowledge” section according to the reviewer’s suggestion.

The text reads: “We used a mixed methods approach to explore the students’ and teachers’ experience when using the IHC primary school resources. Mixed methods research combines quantitative and qualitative methods to answer a research question [30]. This approach helped us deepen our understanding of the phenomenon using different perspectives and methods [31]. Mixed methods research is an evolving methodology that is used in various disciplines, including health science and education [41, 52]; moreover, this approach could be useful to explore health promotion initiatives in schools [53, 54]. In this field, mixed methods can contribute to 1) understand the complexity of health promotion initiatives, 2) identify factors related to efficacy, 3) identify factors related to transferability from one context to another, and 4) point out challenges both from health and educational sectors [55].”

Additional amendments

• We have added a brief appointment about mixed methods integration analysis and findings in “Abstract” and “Discussion; Summary of findings” sections.

• We have included some minor changes in the manuscript to improve the clarity of the text.

• We have updated the list of references.

---

## [Decision Letter · Decision Letter 1]

19 Jun 2023

Piloting the Informed Health Choices resources in Barcelona primary schools: A mixed methods study

PONE-D-23-05069R1

Dear Dr. Martínez García,

We’re pleased to inform you that your manuscript has been judged scientifically suitable for publication and will be formally accepted for publication once it meets all outstanding technical requirements.

Kind regards,

Anastassia Zabrodskaja, Ph.D.

Academic Editor

PLOS ONE

Additional Editor Comments (optional):

Reviewers' comments:

Reviewer's Responses to Questions

**Comments to the Author**

1. If the authors have adequately addressed your comments raised in a previous round of review and you feel that this manuscript is now acceptable for publication, you may indicate that here to bypass the “Comments to the Author” section, enter your conflict of interest statement in the “Confidential to Editor” section, and submit your "Accept" recommendation.

Reviewer #1: All comments have been addressed

Reviewer #3: All comments have been addressed

2. Is the manuscript technically sound, and do the data support the conclusions?

Reviewer #1: Yes

Reviewer #3: Yes

3. Has the statistical analysis been performed appropriately and rigorously? 

Reviewer #1: Yes

Reviewer #3: Yes

4. Have the authors made all data underlying the findings in their manuscript fully available?

Reviewer #1: Yes

Reviewer #3: Yes

5. Is the manuscript presented in an intelligible fashion and written in standard English?

Reviewer #1: Yes

Reviewer #3: Yes

6. Review Comments to the Author

Reviewer #1: This is very interesting research nd I believe the paper was greatly improved. Thanks for dressing all my concerns.

Reviewer #3: All comments have been addressed satisfactorily. I think the manuscript is better and may now be considered worthy of acceptance for publication.

7. PLOS authors have the option to publish the peer review history of their article (what does this mean?). If published, this will include your full peer review and any attached files.

Reviewer #1: No

Reviewer #3: **Yes: **Daniel Semakula

---

## [Editor Report · Acceptance letter]

26 Jun 2023

PONE-D-23-05069R1 

Piloting the Informed Health Choices resources in Barcelona primary schools: A mixed methods study 

Dear Dr. Martínez García:

I'm pleased to inform you that your manuscript has been deemed suitable for publication in PLOS ONE. Congratulations! Your manuscript is now with our production department. 

Kind regards, 

on behalf of

Professor Anastassia Zabrodskaja 

Academic Editor

PLOS ONE